# GLIME: General, Stable and Local LIME Explanation

**Zeren Tan**
Tsinghua University
thutzr1019@gmail.com

**Yang Tian**
Tsinghua University
tyanyang04@gmail.com

**Jian Li**
Tsinghua University
lapordge@gmail.com

## Abstract

As black-box machine learning models grow in complexity and find applications in high-stakes scenarios, it is imperative to provide explanations for their predictions. Although Local Interpretable Model-agnostic Explanations (LIME) [22] is a widely adpoted method for understanding model behaviors, it is unstable with respect to random seeds [35, 24, 3] and exhibits low local fidelity (i.e., how well the explanation approximates the model's local behaviors) [21, 16]. Our study shows that this instability problem stems from small sample weights, leading to the dominance of regularization and slow convergence. Additionally, LIME's sampling neighborhood is non-local and biased towards the reference, resulting in poor local fidelity and sensitivity to reference choice. To tackle these challenges, we introduce GLIME, an enhanced framework extending LIME and unifying several prior methods. Within the GLIME framework, we derive an equivalent formulation of LIME that achieves significantly faster convergence and improved stability. By employing a local and unbiased sampling distribution, GLIME generates explanations with higher local fidelity compared to LIME. GLIME explanations are independent of reference choice. Moreover, GLIME offers users the flexibility to choose a sampling distribution based on their specific scenarios.

## 1 Introduction

Why a patient is predicted to have a brain tumor [10]? Why a credit application is rejected [11]? Why a picture is identified as an electric guitar [22]? As black-box machine learning models continue to evolve in complexity and are employed in critical applications, it is imperative to provide explanations for their predictions, making interpretability a central concern [1]. In response to this imperative, various explanation methods have been proposed [39, 26, 4, 19, 22, 28, 30], aiming to provide insights into the internal mechanisms of deep learning models.

Among the various explanation methods, Local Interpretable Model-agnostic Explanations (LIME) [22] has attracted significant attention, particularly in image classification tasks. LIME explains predictions by assigning each region within an image a weight indicating the influence of this region to the output. This methodology entails segmenting the image into super-pixels, as illustrated in the lower-left portion of Figure 1a, introducing perturbations, and subsequently approximating the local model prediction using a linear model. The approximation is achieved by solving a weighted Ridge regression problem, which estimates the impact (i.e., weight) of each super-pixel on the classifier's output.

Nevertheless, LIME has encountered significant instability due to its random sampling procedure [35, 24, 3]. In LIME, a set of samples perturbing the original image is taken. As illustrated in Figure 1a, LIME explanations generated with two different random seeds display notable disparities, despite using a large sample size (16384). The Jaccard index, measuring similarity between two explanations on a scale from 0 to 1 (with higher values indicating better similarity), is below 0.4. While many prior studies aim to enhance LIME's stability, some sacrifice computational time for stability [24, 40], and others may entail the risk of overfitting [35]. The evident drawback of unstable

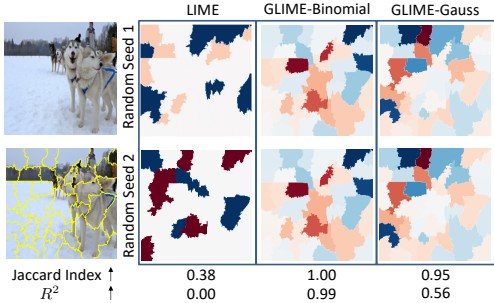
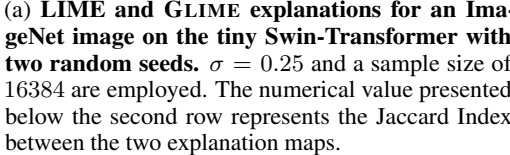

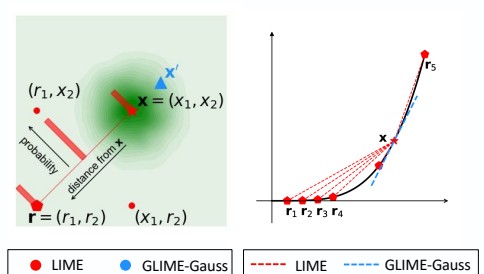

| | LIME | GLIME-Binomial | GLIME-Gauss |
|---|---|---|---|
| Jaccard Index ↑ | 0.38 | 1.00 | 0.95 |
| $R^2$ ↑ | 0.00 | 0.99 | 0.56 |

(a) **LIME and GLIME explanations for an ImageNet image on the tiny Swin-Transformer with two random seeds.** $\sigma = 0.25$ and a sample size of 16384 are employed. The numerical value presented below the second row represents the Jaccard Index between the two explanation maps.

(b) **Distribution of LIME and GLIME-GAUSS in 2D example (left) and 1D example explanations (right).** LIME samples are constrained to one side of $\mathbf{r}$ and are more distant from $\mathbf{x}$. Additionally, LIME explanations exhibit variability with distinct reference points.

Figure 1: **GLIME enhances stability and local fidelity compared to LIME.** (a) LIME demonstrates instability with the default parameter $\sigma$, while GLIME consistently provides meaningful explanations. (b) LIME samples from a biased and non-local neighborhood, a limitation overcome by GLIME.

explanations lies in their potential to mislead end-users and hinder the identification of model bugs and biases, given that LIME explanations lack consistency across different random seeds.

In addition to its inherent instability, LIME has been found to have poor local fidelity [16, 21]. As depicted in Figure 1a, the $R^2$ value for LIME on the sample image approaches zero (refer also to Figure 4b). This problem arises from the non-local and skewed sampling space of LIME, which is biased towards the reference. More precisely, the sampling space of LIME consists of the corner points of the hypercube defined by the explained instance and the selected reference. For instance, in the left section of Figure 1b, only four red points fall within LIME's sampling space, yet these points are distant from $\mathbf{x}$. As illustrated in Figure 3, the $L_2$ distance between LIME samples of the input $\mathbf{x}$ and $\mathbf{x}$ is approximately $0.7\|\mathbf{x}\|_2$ on ImageNet. Although LIME incorporates a weighting function to enforce locality, an explanation cannot be considered as local if the samples themselves are non-local, leading to a lack of local fidelity in the explanation. Moreover, the hypercube exhibits bias towards the reference, resulting in explanations designed to explain only a portion of the local neighborhood. This bias causes LIME to generate different explanations for different references, as illustrated in Figure 1b (refer to Appendix A.4 for more analysis and results).

To tackle these challenges, we present GLIME—a local explanation framework that generalizes LIME and five other methods: KernelSHAP [19], SmoothGrad [28], Gradient [36], DLIME [35], and ALIME [24]. Through a flexible sample distribution design, GLIME produces explanations that are more stable and faithful. Addressing LIME's instability issue, within GLIME, we derive an equivalent form of LIME, denoted as GLIME-BINOMIAL, by integrating the weighting function into the sampling distribution. GLIME-BINOMIAL ensures exponential convergence acceleration compared to LIME when the regularization term is presented. Consequently, GLIME-BINOMIAL demonstrates improved stability compared to LIME while preserving superior local fidelity (see Figure 4). Furthermore, GLIME enhances both local fidelity and stability by sampling from a local distribution independent of any specific reference point.

In summary, our contributions can be outlined as follows:

- We conduct an in-depth analysis to find the source of LIME's instability, revealing the interplay between the weighting function and the regularization term as the primary cause. Additionally, we attribute LIME's suboptimal local fidelity to its non-local and biased sampling space.

- We introduce GLIME as a more general local explanation framework, offering a flexible design for the sampling distribution. With varying sampling distributions and weights, GLIME serves as a generalization of LIME and five other preceding local explanation methods.

- By integrating weights into the sampling distribution, we present a specialized instance of GLIME with a binomial sampling distribution, denoted as GLIME-BINOMIAL. We demonstrate that GLIME-BINOMIAL, while maintaining equivalence to LIME, achieves faster convergence with significantly fewer samples. This indicates that enforcing locality in the sampling distribution is better than using a weighting function.

- With regard to local fidelity, GLIME empowers users to devise explanation methods that exhibit greater local fidelity. This is achieved by selecting a local and unbiased sampling distribution tailored to the specific scenario in which GLIME is applied.

## 2 Preliminary

### 2.1 Notations

Let $\mathcal{X}$ and $\mathcal{Y}$ denote the input and output spaces, respectively, where $\mathcal{X} \subset \mathbb{R}^D$ and $\mathcal{Y} \subset \mathbb{R}$. We specifically consider the scenario in which $\mathcal{X}$ represents the space of images, and $f : \mathcal{X} \to \mathcal{Y}$ serves as a machine learning model accepting an input $\mathbf{x} \in \mathcal{X}$. This study focuses on the classification problem, wherein $f$ produces the probability that the image belongs to a certain class, resulting in $\mathcal{Y} = [0, 1]$.

Before proceeding with explanation computations, a set of features $\{s_i\}_{i=1}^d$ is derived by applying a transformation to $\mathbf{x}$. For instance, $\{s_i\}_{i=1}^d$ could represent image segments (also referred to as super-pixels in LIME) or feature maps obtained from a convolutional neural network. Alternatively, $\{s_i\}_{i=1}^d$ may correspond to raw features, i.e., $\mathbf{x}$ itself. In this context, $\|\cdot\|_0$, $\|\cdot\|_1$, and $\|\cdot\|_2$ denote the $\ell_0$, $\ell_1$, and $\ell_2$ norms, respectively, with $\odot$ representing the element-wise product. Boldface letters are employed to denote vectors and matrices, while non-boldface letters represent scalars or features. $B_{\mathbf{x}}(\epsilon)$ denotes the ball centered at $\mathbf{x}$ with radius $\epsilon$.

### 2.2 A brief introduction to LIME

In this section, we present the original definition and implementation of LIME [22] in the context of image classification. LIME, as a local explanation method, constructs a linear model when provided with an input $\mathbf{x}$ that requires an explanation. The coefficients of this linear model serve as the feature importance explanation for $\mathbf{x}$.

**Features.** For an input $\mathbf{x}$, LIME computes a feature importance vector for the set of features. In the image classification setting, for an image $\mathbf{x}$, LIME initially segments $\mathbf{x}$ into super-pixels $s_1, \ldots, s_d$ using a segmentation algorithm such as Quickshift [32]. Each super-pixel is regarded as a feature for the input $\mathbf{x}$.

**Sample generation.** Subsequently, LIME generates samples within the local vicinity of $\mathbf{x}$ as follows. First, random samples are generated uniformly from $\{0, 1\}^d$. The $j$-th coordinate $z_j'$ for each sample $\mathbf{z}'$ is either 1 or 0, indicating the presence or absence of the super-pixel $s_j$. When $s_j$ is absent, it is replaced by a reference value $r_j$. Common choices for the reference value include a black image, a blurred version of the super-pixel, or the average value of the super-pixel [29, 22, 8]. Then, these $\mathbf{z}'$ samples are transformed into samples in the original input space $\mathbb{R}^D$ by combining them with $\mathbf{x} = (s_1, \ldots, s_d)$ using the element-wise product as follows: $\mathbf{z} = \mathbf{x} \odot \mathbf{z}' + \mathbf{r} \odot (1 - \mathbf{z}')$, where $\mathbf{r}$ is the vector of reference values for each super-pixel, and $\odot$ represents the element-wise product. In other words, $\mathbf{z} \in \mathcal{X}$ is an image that is the same as $\mathbf{x}$, except that those super-pixels $s_j$ with $z_j' = 0$ are replaced by reference values.

**Feature attributions.** For each sample $\mathbf{z}'$ and the corresponding image $\mathbf{z}$, we compute the prediction $f(\mathbf{z})$. Finally, LIME solves the following regression problem to obtain a feature importance vector (also known as feature attributions) for the super-pixels:

$$\mathbf{w}^{\text{LIME}} = \arg\min_{\mathbf{v}} \mathbb{E}_{\mathbf{z}' \sim \text{Uni}(\{0,1\}^d)}[\pi(\mathbf{z}')(f(\mathbf{z}) - \mathbf{v}^\top \mathbf{z}')^2] + \lambda\|\mathbf{v}\|_2^2, \tag{1}$$

where $\mathbf{z} = \mathbf{x} \odot \mathbf{z}' + \mathbf{r} \odot (1 - \mathbf{z}')$, $\pi(\mathbf{z}') = \exp\{-\|\mathbf{1} - \mathbf{z}'\|_2^2/\sigma^2\}$, and $\sigma$ is the kernel width parameter.

**Remark 2.1.** *In practice, we draw samples $\{\mathbf{z}_i'\}_{i=1}^n$ from the uniform distribution $Uni(\{0,1\}^d)$ to estimate the expectation in Equation 1. In the original LIME implementation [22], $\lambda = \alpha/n$ for a*

*constant $\alpha > 0$. This choice has been widely adopted in prior studies [40, 8, 19, 9, 5, 20, 34]. We use $\hat{\mathbf{w}}^{LIME}$ to represent the empirical estimation of $\mathbf{w}^{LIME}$.*

### 2.3 LIME is unstable and has poor local fidelity

**Instability.** To capture the local characteristics of the neighborhood around the input $\mathbf{x}$, LIME utilizes the sample weighting function $\pi(\cdot)$ to assign low weights to samples that exclude numerous super-pixels and, consequently, are located far from $\mathbf{x}$. The parameter $\sigma$ controls the level of locality, with a small $\sigma$ assigning high weights exclusively to samples very close to $\mathbf{x}$ and a large $\sigma$ permitting notable weights for samples farther from $\mathbf{x}$ as well. The default value for $\sigma$ in LIME is $0.25$ for image data. However, as depicted in Figure 1a, LIME demonstrates instability, a phenomenon also noted in prior studies [35, 24, 34]. As showed in Section 4, this instability arises from small $\sigma$ values, leading to very small sample weights and, consequently, slow convergence.

**Poor local fidelity.** LIME also suffers from poor local fidelity [16, 21]. The sampling space of LIME is depicted in Figure 1b. Generally, the samples in LIME exhibit considerable distance from the instance being explained, as illustrated in Figure 3, rendering them non-local. Despite LIME's incorporation of weights to promote locality, it fails to provide accurate explanations for local behaviors when the samples themselves lack local proximity. Moreover, the sampling space of LIME is influenced by the reference, resulting in a biased sampling space and a consequent degradation of local fidelity.

## 3 A general local explanation framework: GLIME

### 3.1 The definition of GLIME

We first present the definition of GLIME and show how it computes the explanation vector $\mathbf{w}^{\text{GLIME}}$. Analogous to LIME, GLIME functions by constructing a model within the neighborhood of the input $\mathbf{x}$, utilizing sampled data from this neighborhood. The coefficients obtained from this model are subsequently employed as the feature importance explanation for $\mathbf{x}$.

**Feature space.** For the provided input $\mathbf{x} \in \mathcal{X} \subset \mathbb{R}^D$, the feature importance explanation is computed for a set of features $\mathbf{s} = (s_1, \ldots, s_d)$ derived from applying a transformation to $\mathbf{x}$. These features $\mathbf{s}$ can represent image segments (referred to as super-pixels in LIME) or feature maps obtained from a convolutional neural network. Alternatively, the features $\mathbf{s}$ can correspond to raw features, i.e., the individual pixels of $\mathbf{x}$. In the context of LIME, the method specifically operates on super-pixels.

**Sample generation.** Given features $\mathbf{s}$, a sample $\mathbf{z}'$ can be generated from the distribution $\mathcal{P}$ defined on the feature space (e.g., $\mathbf{s}$ are super-pixels segmented by a segmentation algorithm such Quickshift [32] and $\mathcal{P} = \text{Uni}(\{0, 1\}^d)$ in LIME). It's important to note that $\mathbf{z}'$ may not belong to $\mathcal{X}$ and cannot be directly input into the model $f$. Consequently, we reconstruct $\mathbf{z} \in \mathbb{R}^D$ in the original input space for each $\mathbf{z}'$ and obtain $f(\mathbf{z})$ (in LIME, a reference $\mathbf{r}$ is first chosen and then $\mathbf{z} = \mathbf{x} \odot \mathbf{z}' + \mathbf{r} \odot (1 - \mathbf{z}')$). Both $\mathbf{z}$ and $\mathbf{z}'$ are then utilized to compute feature attributions.

**Feature attributions.** For each sample $\mathbf{z}'$ and its corresponding $\mathbf{z}$, we compute the prediction $f(\mathbf{z})$. Our aim is to approximate the local behaviors of $f$ around $\mathbf{x}$ using a function $g$ that operates on the feature space. $g$ can take various forms such as a linear model, a decision tree, or any Boolean function operating on Fourier bases [37]. The loss function $\ell(f(\mathbf{z}), g(\mathbf{z}'))$ quantifies the approximation gap for the given sample $\mathbf{z}'$. In the case of LIME, $g(\mathbf{z}') = \mathbf{v}^\top \mathbf{z}'$, and $\ell(f(\mathbf{z}), g(\mathbf{z}')) = (f(\mathbf{z}) - g(\mathbf{z}'))^2$. To derive feature attributions, the following optimization problem is solved:

$$\mathbf{w}^{\text{GLIME}} = \arg\min_{\mathbf{v}} \mathbb{E}_{\mathbf{z}' \sim \mathcal{P}}[\pi(\mathbf{z}')\ell(f(\mathbf{z}), g(\mathbf{z}'))] + \lambda R(\mathbf{v}), \tag{2}$$

where $\pi(\cdot)$ is a weighting function and $R(\cdot)$ serves as a regularization function, e.g., $\| \cdot \|_1$ or $\| \cdot \|_2^2$ (which is used by LIME). We use $\hat{\mathbf{w}}^{\text{GLIME}}$ to represent the empirical estimation of $\mathbf{w}^{\text{GLIME}}$.

**Connection with Existing Frameworks.** Our formulation exhibits similarities with previous frameworks [22, 12]. The generality of GLIME stems from two key aspects: (1) GLIME operates within a broader feature space $\mathbb{R}^d$, in contrast to [22], which is constrained to $\{0, 1\}^d$, and [12], which is confined to raw features in $\mathbb{R}^D$. (2) GLIME can accommodate a more extensive range of distribution choices tailored to specific use cases.

## 3.2 An alternative formulation of GLIME without the weighting function

Indeed, we can readily transform Equation 2 into an equivalent formulation without the weighting function. While this adjustment simplifies the formulation, it also accelerates convergence by sampling from the transformed distribution (see Section 4.1 and Figure 4a). Specifically, we define the transformed sampling distribution as $\widetilde{\mathcal{P}}(\mathbf{z}') = \frac{\pi(\mathbf{z}')\mathcal{P}(\mathbf{z}')}{\int \pi(\mathbf{t})\mathcal{P}(\mathbf{t})d\mathbf{t}}$. Utilizing $\widetilde{\mathcal{P}}$ as the sampling distribution, Equation 2 can be equivalently expressed as

$$\mathbf{w}^{\text{GLIME}} = \arg\min_{\mathbf{v}} \mathbb{E}_{\mathbf{z}' \sim \widetilde{\mathcal{P}}}[\ell(f(\mathbf{z}), g(\mathbf{z}'))] + \frac{\lambda}{Z}R(\mathbf{v}), \quad Z = \mathbb{E}_{\mathbf{t} \sim \mathcal{P}}[\pi(\mathbf{t})\mathcal{P}(\mathbf{t})] \tag{3}$$

It is noteworthy that the feature attributions obtained by solving Equation 3 are equivalent to those obtained by solving Equation 2 (see Appendix B.1 for a formal proof). Therefore, the use of $\pi(\cdot)$ in the formulation is not necessary and can be omitted. Hence, unless otherwise specified, GLIME refers to the framework without the weighting function.

## 3.3 GLIME unifies several previous explanation methods

This section shows how GLIME unifies previous methods. For a comprehensive understanding of the background regarding these methods, kindly refer to Appendix A.6.

**LIME [22] and GLIME-BINOMIAL.** In the case of LIME, it initiates the explanation process by segmenting pixels $x_1, \cdots, x_D$ into super-pixels $s_1, \cdots, s_d$. The binary vector $\mathbf{z}' \sim \mathcal{P} = \text{Uni}(\{0,1\}^d)$ signifies the absence or presence of corresponding super-pixels. Subsequently, $\mathbf{z} = \mathbf{x} \odot \mathbf{z}' + \mathbf{r} \odot (\mathbf{1} - \mathbf{z}')$. The linear model $g(\mathbf{z}') = \mathbf{v}^\top \mathbf{z}'$ is defined on $\{0,1\}^d$. For image explanations, $\ell(f(\mathbf{z}), g(\mathbf{z}')) = (f(\mathbf{z}) - g(\mathbf{z}'))^2$, and the default setting is $\pi(\mathbf{z}') = \exp(-\|\mathbf{1} - \mathbf{z}'\|_0^2/\sigma^2)$, $R(\mathbf{v}) = \|\mathbf{v}\|_2^2$ [22]. Remarkably, LIME is equivalent to the special case GLIME-BINOMIAL without the weighting function (see Appendix B.2 for the formal proof). The sampling distribution of GLIME-BINOMIAL is defined as $\mathcal{P}(\mathbf{z}', \|\mathbf{z}'\|_0 = k) = e^{k/\sigma^2}/(1 + e^{1/\sigma^2})^d$, where $k = 0, 1, \ldots, d$. This distribution is essentially a Binomial distribution. To generate a sample $\mathbf{z}' \in \{0,1\}^d$, one can independently draw $z_i' \in \{0,1\}$ with $\mathbb{P}(z_i = 1) = 1/(1 + e^{-1/\sigma^2})$ for $i = 1, \ldots, d$. The feature importance vector obtained by solving Equation 3 under GLIME-BINOMIAL is denoted as $\mathbf{w}^{\text{Binomial}}$.

**KernelSHAP [19].** In our framework, the formulation of KernelSHAP aligns with that of LIME, with the only difference being $R(\mathbf{v}) = 0$ and $\pi(\mathbf{z}') = (d-1)/(\binom{d}{\|\mathbf{z}'\|_0}\|\mathbf{z}'\|_0(d - \|\mathbf{z}'\|_0))$.

**SmoothGrad [28].** SmoothGrad functions on raw features, specifically pixels in the case of an image. Here, $\mathbf{z} = \mathbf{z}' + \mathbf{x}$, where $\mathbf{z}' \sim \mathcal{N}(\mathbf{0}, \sigma^2\mathbf{I})$. The loss function $\ell(f(\mathbf{z}), g(\mathbf{z}'))$ is represented by the squared loss, while $\pi(\mathbf{z}') = 1$ and $R(\mathbf{v}) = 0$, as established in Appendix B.6.

**Gradient [36].** The Gradient explanation is essentially the limit of SmoothGrad as $\sigma$ approaches 0.

**DLIME [35].** DLIME functions on raw features, where $\mathcal{P}$ is defined over the training data that have the same label with the nearest neighbor of $\mathbf{x}$. The linear model $g(\mathbf{z}') = \mathbf{v}^\top \mathbf{z}'$ is employed with the square loss function $\ell$ and the regularization term $R(\mathbf{v}) = 0$.

**ALIME [24].** ALIME employs an auto-encoder trained on the training data, with its feature space defined as the output space of the auto-encoder. The sample generation process involves introducing Gaussian noise to $\mathbf{x}$. The weighting function in ALIME is denoted as $\pi(\mathbf{z}') = \exp(-\|\mathcal{AE}(\mathbf{x}) - \mathcal{AE}(\mathbf{z}')\|_1)$, where $\mathcal{AE}(\cdot)$ represents the auto-encoder. The squared loss function is chosen as the loss function and no regularization function is applied.

# 4 Stable and locally faithful explanations in GLIME

## 4.1 GLIME-BINOMIAL converges exponentially faster than LIME

To understand the instability of LIME, we demonstrate that the sample weights in LIME are very small, resulting in the domination of the regularization term. Consequently, LIME tends to produce explanations that are close to zero. Additionally, the small weights in LIME lead to a considerably slower convergence compared to GLIME-BINOMIAL, despite both methods converging to the same limit.

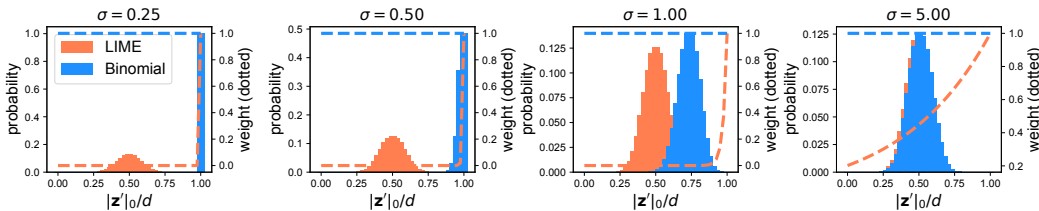

Figure 2: **The distribution and the weight of $\|\mathbf{z}'\|_0$.** In LIME, the distribution of $\|\mathbf{z}'\|_0$ follows a binomial distribution, and it is independent of $\sigma$. In GLIME-BINOMIAL, when $\sigma$ is small, $\|\mathbf{z}'\|_0$ concentrates around $d$, while in LIME, most samples exhibit negligible weights except for the all-one vector $\mathbf{1}$. As $\sigma$ increases, the distributions in GLIME-BINOMIAL converge to those in LIME, and all LIME samples attain non-negligible weights.

**Small sample weights in LIME.** The distribution of the ratio of non-zero elements to the total number of super-pixels, along with the corresponding weights for LIME and GLIME-BINOMIAL, is depicted in Figure 2. Notably, most samples exhibit approximately $d/2$ non-zero elements. However, when $\sigma$ takes values such as 0.25 or 0.5, a significant portion of samples attains weights that are nearly zero. For instance, when $\sigma = 0.25$ and $\|\mathbf{z}'\|_0 = d/2$, $\pi(\mathbf{z}')$ reduces to $\exp(-8d)$, which is approximately $10^{-70}$ for $d = 20$. Even with $\|\mathbf{z}'\|_0 = d - 1$, $\pi(\mathbf{z}')$ equals $e^{-16}$, approximating $10^{-7}$. Since LIME samples from $Uni(\{0, 1\}^d)$, the probability that a sample $\mathbf{z}'$ has $\|\mathbf{z}'\|_0 = d - 1$ or $d$ is approximately $2 \times 10^{-5}$ when $d = 20$. Therefore, most samples have very small weights. Consequently, the sample estimation of the expectation in Equation 1 tends to be much smaller than the true expectation with high probability and is thus inaccurate (see Appendix B.3 for more details). Given the default regularization strength $\lambda = 1$, this imbalance implies the domination of the regularization term in the objective function of Equation 1. As a result, LIME tends to yield explanations close to zero in such cases, diminishing their meaningfulness and leading to instability.

**GLIME converges exponentially faster than LIME in the presence of regularization.** Through the integration of the weighting function into the sampling process, every sample uniformly carries a weight of 1, contributing equally to Equation 3. Our analysis reveals that GLIME requires substantially fewer samples than LIME to transition beyond the regime where the regularization term dominates. Consequently, GLIME-BINOMIAL converges exponentially faster than LIME. Recall that $\hat{\mathbf{w}}^{\text{LIME}}$ and $\hat{\mathbf{w}}^{\text{GLIME}}$ represent the empirical solutions of Equation 1 and Equation 3, respectively, obtained by replacing the expectations with the sample average. $\hat{\mathbf{w}}^{\text{BINOMIAL}}$ is the empirical solution of Equation 3 with the transformed sampling distribution $\widetilde{\mathcal{P}}(\mathbf{z}', \|\mathbf{z}'\|_0 = k) = e^{k/\sigma^2}/(1 + e^{1/\sigma^2})^d$, where $k = 0, 1, \cdots, d$. In the subsequent theorem, we present the sample complexity bound for LIME (refer to Appendix B.4 for proof).

**Theorem 4.1.** *Suppose samples $\{\mathbf{z}_i'\}_{i=1}^n \sim Uni(\{0, 1\}^d)$ are used to compute the LIME explanation. For any $\epsilon > 0, \delta \in (0, 1)$, if $n = \Omega(\epsilon^{-2}d^9 2^{8d} e^{8/\sigma^2} \log(4d/\delta)), \lambda \leq n$, we have $\mathbb{P}(\|\hat{\mathbf{w}}^{\text{LIME}} - \mathbf{w}^{\text{LIME}}\|_2 < \epsilon) \geq 1 - \delta$. $\mathbf{w}^{\text{LIME}} = \lim_{n \to \infty} \hat{\mathbf{w}}^{\text{LIME}}$.*

Next, we present the sample complexity bound for GLIME (refer to Appendix B.5 for proof).

**Theorem 4.2.** *Suppose $\mathbf{z}' \sim \mathcal{P}$ such that the largest eigenvalue of $\mathbf{z}'(\mathbf{z}')^\top$ is bounded by $R$ and $\mathbb{E}[\mathbf{z}'(\mathbf{z}')^\top] = (\alpha_1 - \alpha_2)\mathbf{I} + \alpha_2 \mathbf{1}\mathbf{1}^\top, \|Var(\mathbf{z}'(\mathbf{z}')^\top)\|_2 \leq \nu^2, |(\mathbf{z}'f(\mathbf{z}))_i| \leq M$ for some $M > 0$. $\{\mathbf{z}_i'\}_{i=1}^n$ are i.i.d. samples from $\mathcal{P}$ and are used to compute GLIME explanation $\hat{\mathbf{w}}^{\text{GLIME}}$. For any $\epsilon > 0, \delta \in (0, 1)$, if $n = \Omega(\epsilon^{-2}M^2\nu^2 d^3 \gamma^4 \log(4d/\delta))$ where $\gamma$ is a function of $\lambda, d, \alpha_1, \alpha_2$, we have $\mathbb{P}(\|\hat{\mathbf{w}}^{\text{GLIME}} - \mathbf{w}^{\text{GLIME}}\|_2 < \epsilon) \geq 1 - \delta$. $\mathbf{w}^{\text{GLIME}} = \lim_{n \to \infty} \hat{\mathbf{w}}^{\text{GLIME}}$.*

Since GLIME-BINOMIAL samples from a binomial distribution, which is sub-Gaussian with parameters $M = \sqrt{d}, \nu = 2, \alpha_1 = 1/(1 + e^{-1/\sigma^2}), \alpha_2 = 1/(1 + e^{-1/\sigma^2})^2$, and $\gamma(\alpha_1, \alpha_2, d) = de^{2/\sigma^2}$ (refer to Appendix B.5 for proof), we derive the following corollary:

**Corollary 4.3.** *Suppose $\{\mathbf{z}_i'\}_{i=1}^n$ are i.i.d. samples from $\widetilde{\mathcal{P}}(\mathbf{z}', \|\mathbf{z}'\|_0 = k) = e^{k/\sigma^2}/(1 + e^{1/\sigma^2})^d, k = 1, \ldots, d$ and are used to compute GLIME-BINOMIAL explanation. For any $\epsilon > 0, \delta \in (0, 1)$, if $n = \Omega(\epsilon^{-2}d^5 e^{4/\sigma^2} \log(4d/\delta))$, we have $\mathbb{P}(\|\hat{\mathbf{w}}^{\text{Binomial}} - \mathbf{w}^{\text{Binomial}}\|_2 < \epsilon) \geq 1 - \delta$. $\mathbf{w}^{\text{Binomial}} = \lim_{n \to \infty} \hat{\mathbf{w}}^{\text{Binomial}}$.*

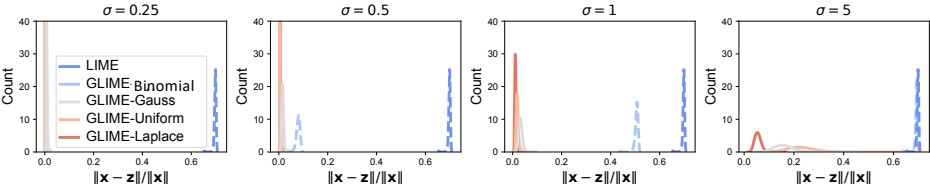

Figure 3: **The distribution of sample distances to the original input.** The samples produced by LIME display a considerable distance from the original input $\mathbf{x}$, whereas the samples generated by GLIME demonstrate a more localized distribution. LIME has a tendency to overlook sampling points that are in close proximity to $\mathbf{x}$.

Comparing the sample complexities outlined in Theorem 4.1 and Corollary 4.3, it becomes evident that LIME necessitates an exponential increase of $\exp(d, \sigma^{-2})$ more samples than GLIME-BINOMIAL for convergence. Despite both LIME and GLIME-BINOMIAL samples being defined on the binary set $\{0, 1\}$, the weight $\pi(\mathbf{z}')$ associated with a sample $\mathbf{z}'$ in LIME is notably small. Consequently, the square loss term in LIME is significantly diminished compared to that in GLIME-BINOMIAL. This situation results in the domination of the regularization term over the square loss term, leading to solutions that are close to zero. For stable solutions, it is crucial that the square loss term is comparable to the regularization term. Consequently, GLIME-BINOMIAL requires significantly fewer samples than LIME to achieve stability.

## 4.2 Designing locally faithful explanation methods within GLIME

**Non-local and biased sampling in LIME.** LIME employs uniform sampling from $\{0, 1\}^d$ and subsequently maps the samples to the original input space $\mathcal{X}$ with the inclusion of a reference. Despite the integration of a weighting function to enhance locality, the samples $\{\mathbf{z}_i\}_{i=1}^n$ generated by LIME often exhibit non-local characteristics, limiting their efficacy in capturing the local behaviors of the model $f$ (as depicted in Figure 3). This observation aligns with findings in [16, 21], which demonstrate that LIME frequently approximates the global behaviors instead of the local behaviors of $f$. As illustrated earlier, the weighting function contributes to LIME's instability, emphasizing the need for explicit enforcement of locality in the sampling process.

**Local and unbiased sampling in GLIME.** In response to these challenges, GLIME introduces a sampling procedure that systematically enforces locality without reliance on a reference point. One approach involves sampling $\mathbf{z}' \sim \mathcal{P} = \mathcal{N}(\mathbf{0}, \sigma^2 \mathbf{I})$ and subsequently obtaining $\mathbf{z} = \mathbf{x} + \mathbf{z}'$. This method, referred to as GLIME-GAUSS, utilizes a weighting function $\pi(\cdot) \equiv 1$, with other components chosen to mirror those of LIME. The feature attributions derived from this approach successfully mitigate the aforementioned issues. Similarly, alternative distributions, such as $\mathcal{P} = \text{Laplace}(\mathbf{0}, \sigma)$ or $\mathcal{P} = \text{Uni}([-\sigma, \sigma]^d)$, can be employed, resulting in explanation methods known as GLIME-LAPLACE and GLIME-UNIFORM, respectively.

## 4.3 Sampling distribution selection for user-specific objectives

Users may possess specific objectives they wish the explanation method to fulfill. For instance, if a user seeks to enhance local fidelity within a neighborhood of radius $\epsilon$, they can choose a distribution and corresponding parameters aligned with this objective (as depicted in Figure 5). The flexible design of the sample distribution in GLIME empowers users to opt for a distribution that aligns with their particular use cases. Furthermore, within the GLIME framework, it is feasible to integrate feature correlation into the sampling distribution, providing enhanced flexibility. In summary, GLIME affords users the capability to make more tailored choices based on their individual needs and objectives.

# 5 Experiments

**Dataset and models.** Our experiments are conducted on the ImageNet dataset[1]. Specifically, we randomly choose 100 classes and select an image at random from each class. The models chosen for explanation are ResNet18 [13] and the tiny Swin-Transformer [18] (refer to Appendix A.7 for results). Our implementation is derived from the official implementation of LIME[2]. The default segmentation algorithm in LIME, Quickshift [32], is employed. Implementation details of our experiments are provided in Appendix A.1. For experiment results on text data, please refer to Appendix A.9. For experiment results on ALIME, please refer to Appendix A.8.

**Metrics.** (1) *Stability*: To gauge the stability of an explanation method, we calculate the average top-$K$ Jaccard Index (JI) for explanations generated by 10 different random seeds. Let $\mathbf{w}_1, \ldots, \mathbf{w}_{10}$ denote the explanations obtained from 10 random seeds. The indices corresponding to the top-$K$ largest values in $\mathbf{w}_i$ are denoted as $R_{i,:K}$. The average Jaccard Index between pairs of $R_{i,:K}$ and $R_{j,:K}$ is then computed, where $\mathrm{JI}(A, B) = |A \cap B|/|A \cup B|$.

(2) *Local Fidelity*: To evaluate the local fidelity of explanations, reflecting how well they capture the local behaviors of the model, we employ two approaches. For LIME, which uses a non-local sampling neighborhood, we use the $R^2$ score returned by the LIME implementation for local fidelity assessment [33]. Within GLIME, we generate samples $\{\mathbf{z}_i\}_{i=1}^m$ and $\{\mathbf{z}_i'\}_{i=1}^m$ from the neighborhood $B_{\mathbf{x}}(\epsilon)$. The squared difference between the model's output and the explanation's output on these samples is computed. Specifically, for a sample $\mathbf{z}$, we calculate $(f(\mathbf{z}) - \hat{\mathbf{w}}^\top \mathbf{z}')^2$ for the explanation $\hat{\mathbf{w}}$. The local fidelity of an explanation $\hat{\mathbf{w}}$ at the input $\mathbf{x}$ is defined as $1/(1 + \frac{1}{m}\sum_i (f(\mathbf{z}_i) - \hat{\mathbf{w}}^\top \mathbf{z}_i')^2)$, following the definition in [16]. To ensure a fair comparison between different distributions in GLIME, we set the variance parameter of each distribution to match that of the Gaussian distribution. For instance, when sampling from the Laplace distribution, we use Laplace$(\mathbf{0}, \sigma/\sqrt{2})$, and when sampling from the uniform distribution, we use Uni$([-\sqrt{3}\sigma, \sqrt{3}\sigma]^d)$.

## 5.1 Stability of LIME and GLIME

**LIME's instability and the influence of regularization/weighting.** In Figure 4a, it is evident that LIME without the weighting function (LIME + $\pi = 1$) demonstrates greater stability compared to its weighted counterpart, especially when $\sigma$ is small (e.g., $\sigma = 0.25, 0.5$). This implies that the weighting function contributes to instability in LIME. Additionally, we observe that LIME without regularization (LIME + $\lambda = 0$) exhibits higher stability than the regularized LIME, although the improvement is not substantial. This is because, when $\sigma$ is small, the sample weights approach zero, causing the Ridge regression problem to become low-rank, leading to unstable solutions. Conversely, when $\sigma$ is large, significant weights are assigned to all samples, reducing the effectiveness of regularization. For instance, when $\sigma = 5$ and $d = 40$, most samples carry weights around 0.45, and even samples with only one non-zero element left possess weights of approximately 0.2. In such scenarios, the regularization term does not dominate, even with limited samples. This observation is substantiated by the comparable performance of LIME, LIME+$\pi = 1$, and LIME+$\lambda = 0$ when $\sigma = 1$ and 5. Further results are presented in Appendix A.2.

**Enhancing stability in LIME with GLIME.** In Figure 4a, it is evident that LIME achieves a Jaccard Index of approximately 0.4 even with over 2000 samples when using the default $\sigma = 0.25$. In contrast, both GLIME-BINOMIAL and GLIME-GAUSS provide stable explanations with only 200-400 samples. Moreover, with an increase in the value of $\sigma$, the convergence speed of LIME also improves. However, GLIME-BINOMIAL consistently outperforms LIME, requiring fewer samples for comparable stability. The logarithmic scale of the horizontal axis in Figure 4a highlights the exponential faster convergence of GLIME compared to LIME.

**Convergence of LIME and GLIME-BINOMIAL to a common limit.** In Figure 8 of Appendix A.3, we explore the difference and correlation between explanations generated by LIME and GLIME-BINOMIAL. Mean Squared Error (MSE) and Mean Absolute Error (MAE) are employed as metrics to quantify the dissimilarity between the explanations, while Pearson correlation and Spearman rank correlation assess their degree of correlation. As the sample size increases, both LIME and GLIME-

---

[1]Code is available at `https://github.com/thutzr/GLIME-General-Stable-and-Local-LIME-Explanation`
[2]`https://github.com/marcotcr/lime`

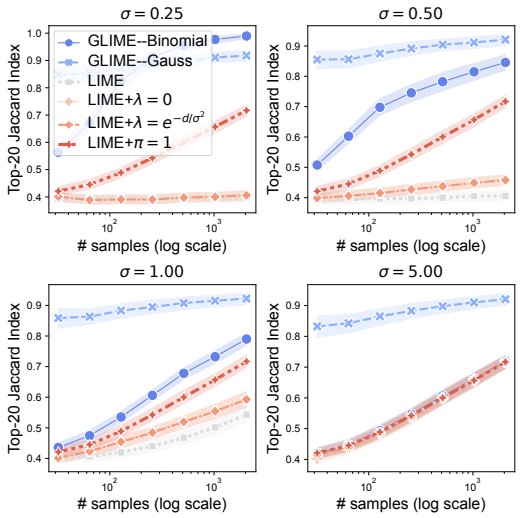

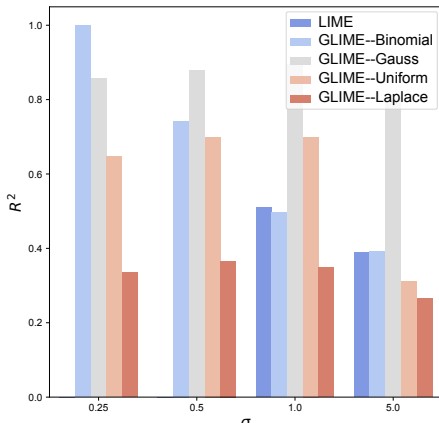

(a) **Stability of various methods.** The reported metric is the Top-20 Jaccard index. Instances of LIME with no regularization and no weighting are denoted as LIME+$\lambda = 0$ and LIME+$\pi = 1$, respectively. LIME exhibits instability when $\sigma$ is small, whereas GLIME demonstrates enhanced stability across different $\sigma$ values. Notably, LIME achieves greater stability without weighting or regularization when $\sigma$ is small. Conversely, regularization and weighting have minimal impact on the stability of LIME when $\sigma$ is large.

(b) $R^2$ **comparison between LIME and various GLIME methods with different sampling distributions.** We utilize 2048 samples to compute explanations and the corresponding $R^2$ for each image and each method. LIME exhibits nearly zero $R^2$ when $\sigma = 0.25$ and $0.5$, suggesting minimal explanatory power. In general, the $R^2$ of LIME is consistently lower than that of GLIME, underscoring GLIME's enhancement in local fidelity.

Figure 4: GLIME consistently enhances stability and local fidelity compared to LIME across various values of $\sigma$.

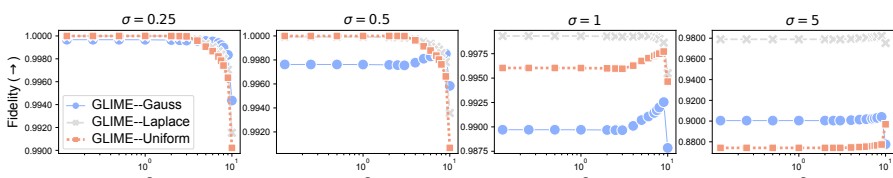

Figure 5: **Local fidelity of GLIME across different neighborhood radii.** Explanations produced under a distribution with a standard deviation of $\sigma$ demonstrate the ability to capture behaviors within local neighborhoods with radii exceeding $\sigma$.

BINOMIAL exhibit greater similarity and higher correlation. The dissimilarity in their explanations diminishes rapidly, approaching zero when $\sigma$ is significantly large (e.g., $\sigma = 5$).

## 5.2 Local fidelity of LIME and GLIME

**Enhancing local fidelity with GLIME.** A comparison of the local fidelity between LIME and the explanation methods generated by GLIME is presented in Figure 4b. Utilizing 2048 samples for each image to compute the $R^2$ score, GLIME consistently demonstrates superior local fidelity compared to LIME. Particularly, when $\sigma = 0.25$ and $0.5$, LIME exhibits local fidelity that is close to zero, signifying that the linear approximation model $(\hat{\mathbf{w}}^{\text{LIME}})^\top \mathbf{z}'$ is nearly constant. Through the explicit integration of locality into the sampling process, GLIME significantly improves the local fidelity of the explanations.

**Local fidelity analysis of GLIME under various sampling distributions.** In Figure 5, we assess the local fidelity of GLIME employing diverse sampling distributions: $\mathcal{N}(\mathbf{0}, \sigma^2\mathbf{I})$, Laplace$(\mathbf{0}, \sigma/\sqrt{2})$,

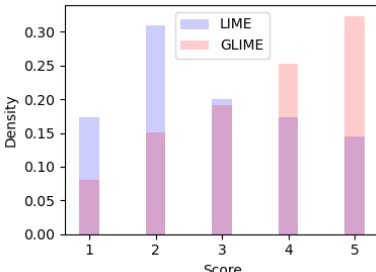 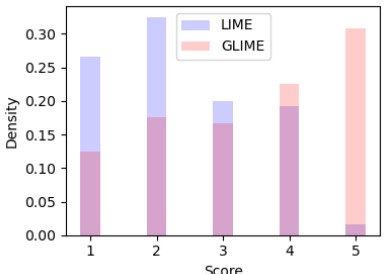

(a) **Human-interpretability results for accurate predictions.** Participants consistently rated GLIME significantly higher than LIME, indicating that GLIME serves as a more effective tool for explaining model predictions.

(b) **Human-interpretability results for incorrect predictions.** Participants consistently rated GLIME significantly higher than LIME, indicating that GLIME excels in identifying the model's errors more effectively than LIME.

Figure 6: GLIME helpes explaining model predictions better than LIME.

and $\mathrm{Uni}([-\sqrt{3}\sigma, \sqrt{3}\sigma]^d)$. The title of each sub-figure corresponds to the standard deviation of these distributions. Notably, we observe that the value of $\sigma$ does not precisely align with the radius $\epsilon$ of the intended local neighborhood for explanation. Instead, local fidelity tends to peak at larger $\epsilon$ values than the corresponding $\sigma$. Moreover, different sampling distributions achieve optimal local fidelity for distinct $\epsilon$ values. This underscores the significance of selecting an appropriate distribution and parameter values based on the specific radius $\epsilon$ of the local neighborhood requiring an explanation. Unlike LIME, GLIME provides the flexibility to accommodate such choices. For additional results and analysis, please refer to Appendix A.5.

## 5.3 Human experiments

In addition to numerical experiments, we conducted human-interpretability experiments to evaluate whether GLIME provides more meaningful explanations to end-users. The experiments consist of two parts, with 10 participants involved in each. The details of the procedures employed in conducting the experiments is presented in the following.

1. Can GLIME improve the comprehension of the model's predictions? To assess this, we choose images for which the model's predictions are accurate. Participants are presented with the original images, accompanied by explanations generated by both LIME and GLIME. They are then asked to evaluate the degree of alignment between the explanations from these algorithms and their intuitive understanding. Using a 1-5 scale, where 1 indicates a significant mismatch and 5 signifies a strong correspondence, participants rate the level of agreement.

2. Can GLIME assist in identifying the model's errors? To explore this, we select images for which the model's predictions are incorrect. Participants receive the original images along with explanations generated by both LIME and GLIME. They are then asked to assess the degree to which these explanations aid in understanding the model's behaviors and uncovering the reasons behind the inaccurate predictions. Using a 1-5 scale, where 1 indicates no assistance and 5 signifies substantial aid, participants rate the level of support provided by the explanations.

Figure 6 presents the experimental results. When participants examined images with accurate model predictions, along with explanations from LIME and GLIME, they assigned an average score of 2.96 to LIME and 3.37 to GLIME. On average, GLIME received a score 0.41 higher than LIME. Notably, in seven out of the ten instances, GLIME achieved a higher average score than LIME.

In contrast, when participants examined images with incorrect model predictions, accompanied by explanations from LIME and GLIME, they assigned an average score of 2.33 to LIME and 3.42 to GLIME. Notably, GLIME outperformed LIME with an average score 1.09 higher across all ten images. These results strongly indicate that GLIME excels in explaining the model's behaviors.

# 6 Related work

**Post-hoc local explanation methods.** In contrast to inherently interpretable models, black-box models can be explained through post-hoc explanation methods, which are broadly categorized as model-agnostic or model-specific. Model-specific approaches, such as Gradient [2, 27], SmoothGrad [28], and Integrated Gradient [30], assume that the explained model is differentiable and that gradient access is available. For instance, SmoothGrad generates samples from a Gaussian distribution centered at the given input and computes their average gradient to mitigate noise. On the other hand, model-agnostic methods, including LIME [22] and Anchor [23], aim to approximate the local model behaviors using interpretable models, such as linear models or rule lists. Another widely-used model-agnostic method, SHAP [19], provides a unified framework that computes feature attributions based on the Shapley value and adheres to several axioms.

**Instability of LIME.** Despite being widely employed, LIME is known to be unstable, evidenced by divergent explanations under different random seeds [35, 34, 38]. Many efforts have been devoted to stabilize LIME explanations. Zafar et al. [35] introduced a deterministic algorithm that utilizes hierarchical clustering for grouping training data and k-nearest neighbors for selecting relevant data samples. However, the resulting explanations may not be a good local approximation. Addressing this concern, Shankaranarayana et al. [24] trained an auto-encoder to function as a more suitable weighting function in LIME. Shi et al. [25] incorporated feature correlation into the sampling step and considered a more restricted sampling distribution, thereby enhancing stability. Zhou et al. [40] employed a hypothesis testing framework to determine the necessary number of samples for ensuring stable explanations. However, this improvement came at the expense of a substantial increase in computation time.

**Impact of references.** LIME, along with various other explanation methods, relies on references (also known as baseline inputs) to generate samples. References serve as uninformative inputs meant to represent the absence of features [4, 30, 26]. Choosing an inappropriate reference can lead to misleading explanations. For instance, if a black image is selected as the reference, important black pixels may not be highlighted [15, 6]. The challenge lies in determining the appropriate reference, as different types of references may yield different explanations [14, 6, 15]. In [15], both black and white references are utilized, while [7] employs constant, noisy, and Gaussian blur references simultaneously. To address the reference specification issue, [6] proposes Expected Gradient, considering each instance in the data distribution as a reference and averaging explanations computed across all references.

# 7 Conclusion

In this paper, we introduce GLIME, a novel framework that extends the LIME method for local feature importance explanations. By explicitly incorporating locality into the sampling procedure and enabling more flexible distribution choices, GLIME mitigates the limitations of LIME, such as instability and low local fidelity. Experimental results on ImageNet data demonstrate that GLIME significantly enhances stability and local fidelity compared to LIME. While our experiments primarily focus on image data, the applicability of our approach readily extends to text and tabular data.

# 8 Acknowledgement

The authors would like to thank the anonymous reviewers for their constructive comments. Zeren Tan and Jian Li are supported by the National Natural Science Foundation of China Grant (62161146004). Yang Tian is supported by the Artificial and General Intelligence Research Program of Guo Qiang Research Institute at Tsinghua University (2020GQG1017).

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

# A More discussions

## A.1 Implementation details

**Dataset selection.** The experiments use images from the validation set of the ImageNet-1k dataset. To ensure consistency, a fixed random seed (2022) is employed. Specifically, 100 classes are uniformly chosen at random, and for each class, an image is randomly selected.

**Models.** The pretrained models used are sourced from `torchvision.models`, with the `weights` parameter set to `IMAGENET1K_V1`.

**Feature transformation.** The initial step involves cropping each image to dimensions of (224, 224, 3). The `quickshift` method from `scikit-image` is then employed to segment images into super-pixels, with specific parameters set as follows: `kernel_size=4`, `max_dist=200`, `ratio=0.2`, and `random_seed=2023`. This approach aligns with the default setting in LIME, except for the modified fixed random seed. Consistency in the random seed ensures that identical images result in the same super-pixels, thereby isolating the source of instability to the calculation of explanations. However, for different images, they are still segmented in different ways.

**Computing explanations.** The implemented procedure follows the original setup in LIME. The `hide_color` parameter is configured as `None`, causing the average value of each super-pixel to act as its reference when the super-pixel is removed. The `distance_metric` is explicitly set to `l2`, as recommended for image data in LIME [22]. The default value for `alpha` in Ridge regression is 1, unless otherwise specified. For each image, the model $f$ infers the most probable label, and the explanation pertaining to that label is computed. Ten different random seeds are utilized to compute explanations for each image. The `random_seed` parameter in both the `LimeImageExplainer` and the `explain_instance` function is set to these specific random seeds.

## A.2 Stability of LIME and GLIME

Figure 7 illustrates the top-1, top-5, top-10, and average Jaccard indices. Importantly, the results presented in Figure 7 closely align with those in Figure 4a. In summary, it is evident that GLIME consistently provides more stable explanations compared to LIME.

## A.3 LIME and GLIME-BINOMIAL converge to the same limit

In Figure 8, the difference and correlation between explanations generated by LIME and GLIME-BINOMIAL are presented. With an increasing sample size, the explanations from LIME and GLIME-BINOMIAL become more similar and correlated. The difference between their explanations rapidly converges to zero, particularly when $\sigma$ is large, such as in the case of $\sigma = 5$. While LIME exhibits a slower convergence, especially with small $\sigma$, it is impractical to continue sampling until their difference fully converges. Nevertheless, the correlation between LIME and GLIME-BINOMIAL strengthens with an increasing number of samples, indicating their convergence to the same limit as the sample size grows.

## A.4 LIME explanations are different for different references.

The earlier work by Jain et al. [14] has underscored the instability of LIME regarding references. As shown in Section 4.2, this instability originates from LIME's sampling distribution, which relies on the chosen reference $\mathbf{r}$. Additional empirical evidence is presented in Figure 9. Six distinct references—black, white, red, blue, yellow image, and the average value of the removed super-pixel (the default setting for LIME)—are selected. The average Jaccard indices for explanations computed using these various references are detailed in Figure 9. The results underscore the sensitivity of LIME to different references.

Different references result in LIME identifying distinct features as the most influential, even with a sample size surpassing 2000. Particularly noteworthy is that, with a sample size exceeding 2000, the top-1 Jaccard index consistently remains below 0.7, underscoring LIME's sensitivity to reference variations.

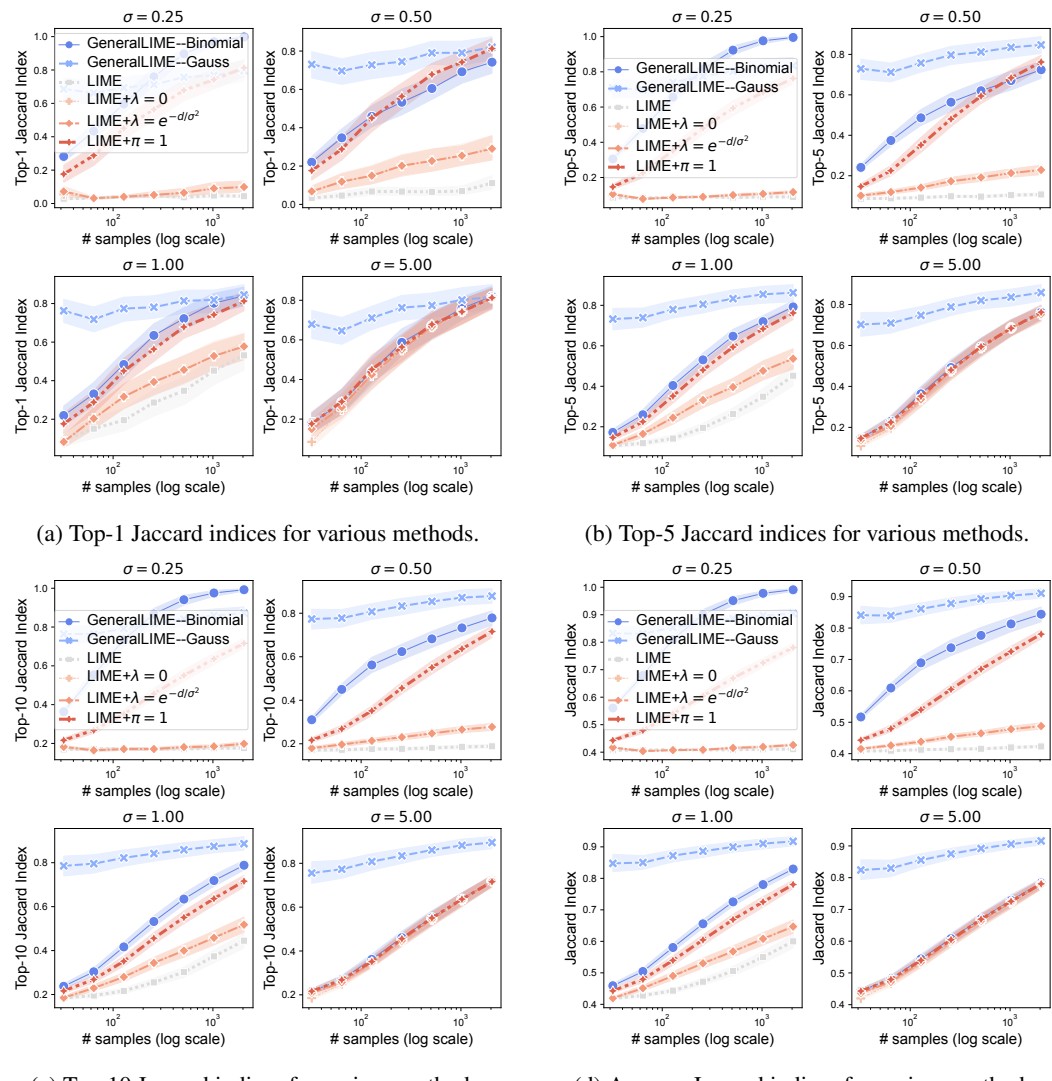

(a) Top-1 Jaccard indices for various methods.

(b) Top-5 Jaccard indices for various methods.

(c) Top-10 Jaccard indices for various methods.

(d) Average Jaccard indices for various methods.

Figure 7: Top-1, 5, 10, and average Jaccard indices are computed for various methods. The average Jaccard index is obtained by averaging the top-1 to top-d Jaccard indices.

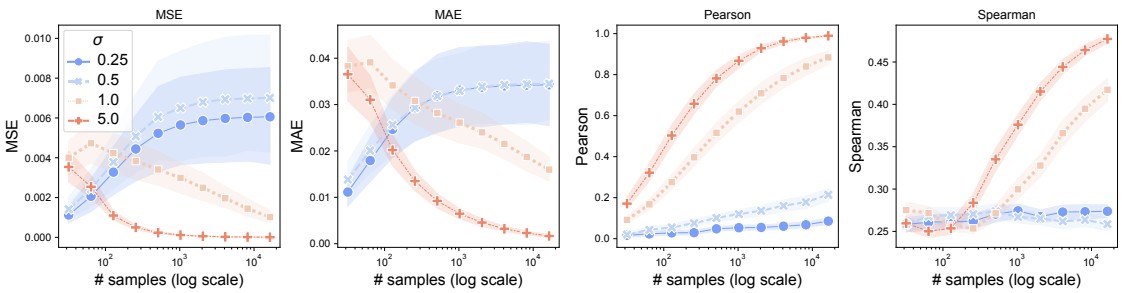

Figure 8: **Difference and correlation between LIME and GLIME-BINOMIAL explanations.** Mean Squared Error (MSE) and Mean Absolute Error (MAE) serve as metrics to evaluate the dissimilarity between explanations provided by LIME and GLIME-BINOMIAL. Pearson and Spearman correlation coefficients are employed to quantify the correlation between these explanations. With an increasing number of samples, the explanations from LIME and GLIME-BINOMIAL tend to show greater similarity. Notably, the dissimilarity and correlation of explanations between LIME and GLIME-BINOMIAL converge more rapidly when $\sigma$ is higher.

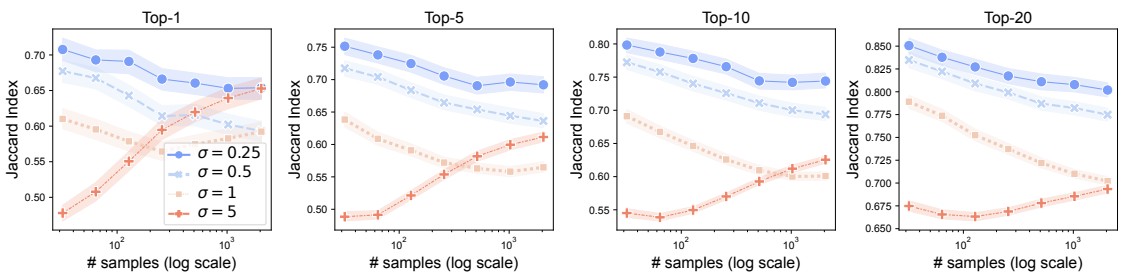

Figure 9: The Top-$K$ Jaccard index of explanations, computed with different references, consistently stays below 0.7, even when the sample size exceeds 2000.

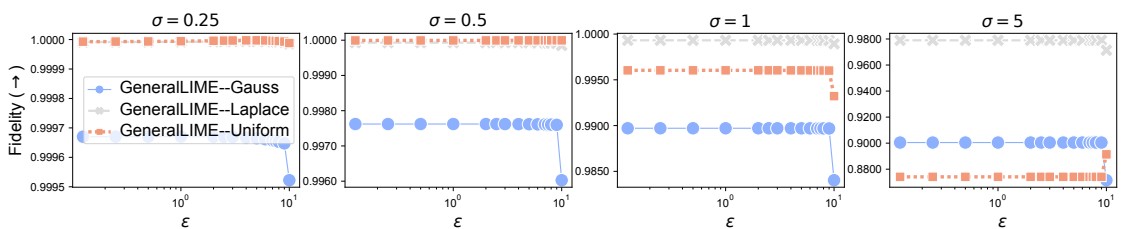

Figure 10: The local fidelity of GLIME in the $\ell_1$ neighborhood

## A.5 The local fidelity of GLIME

Figure 5 presents the local fidelity of GLIME, showcasing samples from the $\ell_2$ neighborhood $\{\mathbf{z}|\|\mathbf{z} - \mathbf{x}\|_2 \leq \epsilon\}$ around $\mathbf{x}$. Additionally, Figure 10 and Figure 11 illustrate the local fidelity of GLIME within the $\ell_1$ neighborhood $\{\mathbf{z}|\|\mathbf{z} - \mathbf{x}\|_1 \leq \epsilon\}$ and the $\ell_\infty$ neighborhood $\{\mathbf{z}|\|\mathbf{z} - \mathbf{x}\|_\infty \leq \epsilon\}$, respectively.

A comparison between Figure 5 and Figure 10 reveals that, for the same $\sigma$, GLIME can explain the local behaviors of $f$ within a larger radius in the $\ell_1$ neighborhood compared to the $\ell_2$ neighborhood. This difference arises from the fact that $\{\mathbf{z}|\|\mathbf{z} - \mathbf{x}\|_2 \leq \epsilon\}$ defines a larger neighborhood compared to $\{\mathbf{z}|\|\mathbf{z} - \mathbf{x}\|_1 \leq \epsilon\}$ with the same radius $\epsilon$.

Likewise, the set $\{\mathbf{z}|\|\mathbf{z} - \mathbf{x}\|_\infty \leq \epsilon\}$ denotes a larger neighborhood than $\{\mathbf{z}|\|\mathbf{z} - \mathbf{x}\|_2 \leq \epsilon\}$, causing the local fidelity to peak at a smaller radius $\epsilon$ for the $\ell_\infty$ neighborhood compared to the $\ell_2$ neighborhood under the same $\sigma$.

Remarkably, GLIME-LAPLACE consistently demonstrates superior local fidelity compared to GLIME-GAUSS and GLIME-UNIFORM. Nevertheless, in cases with larger $\epsilon$, GLIME-GAUSS sometimes surpasses the others. This observation implies that the choice of sampling distribution should be contingent on the particular radius of the local neighborhood intended for explanation by the user.

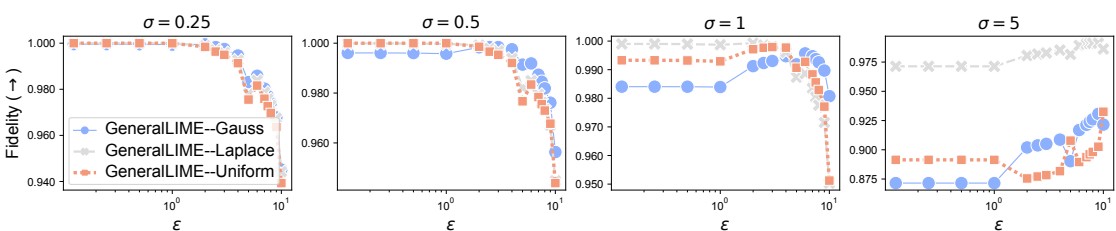

Figure 11: The local fidelity of GLIME in the $\ell_\infty$ neighborhood

## A.6 GLIME unifies several previous methods

**KernelSHAP [19].** KernelSHAP integrates the principles of LIME and Shapley values. While LIME employs a linear explanation model to locally approximate $f$, KernelSHAP seeks a linear explanation model that adheres to the axioms of Shapley values, including local accuracy, missingness, and consistency [19]. Achieving this involves careful selection of the loss function $\ell(\cdot,\cdot)$, the weighting function $\pi(\cdot)$, and the regularization term $R$. The choices for these parameters in LIME often violate local accuracy and/or consistency, unlike the selections made in KernelSHAP, which are proven to adhere to these axioms (refer to Theorem 2 in [19]).

**Gradient [2, 27].** This method computes the gradient $\nabla f$ to assess the impact of each feature under infinitesimal perturbation [2, 27].

**SmoothGrad [28].** Acknowledging that standard gradient explanations may contain noise, Smooth-Grad introduces a method to alleviate noise by averaging gradients within the local neighborhood of the explained input [28]. Consequently, the feature importance scores are computed as $\mathbb{E}_{\epsilon \sim \mathcal{N}(0,\sigma^2 \mathbf{I})}[\nabla f(\mathbf{x} + \epsilon)]$.

**DLIME [35].** Diverging from random sampling, DLIME seeks a deterministic approach to sample acquisition. In its process, DLIME employs agglomerative Hierarchical Clustering to group the training data, and subsequently utilizes k-Nearest Neighbour to select the cluster corresponding to the explained instance. The DLIME explanation is then derived by constructing a linear model based on the data points within the identified cluster.

**ALIME [24]:** ALIME leverages an auto-encoder to assign weights to samples. Initially, an auto-encoder, denoted as $\mathcal{AE}(\cdot)$, is trained on the training data. Subsequently, the method involves sampling $n$ nearest points to $\mathbf{x}$ from the training dataset. The distances between these samples and the explained instance $\mathbf{x}$ are assessed using the $\ell_1$ distance between their embeddings, obtained through the application of the auto-encoder $\mathcal{AE}(\cdot)$. For a sample $\mathbf{z}$, its distance from $\mathbf{x}$ is measured as $\|\mathcal{AE}(\mathbf{z}) - \mathcal{AE}(\mathbf{x})\|_1$, and its weight is computed as $\exp(-\|\mathcal{AE}(\mathbf{z}) - \mathcal{AE}(\mathbf{x})\|_1)$. The final explanation is derived by solving a weighted Ridge regression problem.

## A.7 Results on tiny Swin-Transformer [18]

The findings on the tiny Swin-Transformer align with those on ResNet18, providing additional confirmation that GLIME enhances stability and local fidelity compared to LIME. Please refer to Figure 12, Figure 13 and Figure 14 for results.

## A.8 Comparing GLIME with ALIME

While ALIME [24] improves upon the stability and local fidelity of LIME, GLIME consistently surpasses ALIME. A key difference between ALIME and LIME lies in their methodologies: ALIME employs an encoder to transform samples into an embedding space, calculating their distance from the input to be explained as $\|\mathcal{AE}(\mathbf{z}) - \mathcal{AE}(\mathbf{x})\|_1$, whereas LIME utilizes a binary vector $\mathbf{z} \in \{0,1\}^d$ to represent a sample, measuring the distance from the explained input as $\|\mathbf{1} - \mathbf{z}\|_2$.

Because ALIME relies on distance in the embedding space to assign weights to samples, there is a risk of generating very small sample weights if the produced samples are far from $\mathbf{x}$, potentially resulting in instability issues.

In our ImageNet experiments comparing GLIME and ALIME, we utilize the VGG16 model from the repository `imagenet-autoencoder`[3] as the encoder in ALIME. The outcomes of these experiments are detailed in Table 1. The findings demonstrate that, although ALIME demonstrates enhanced stability compared to LIME, this improvement is not as substantial as the improvement achieved by GLIME, particularly under conditions of small $\sigma$ or sample size.

## A.9 Experiment results on IMDb

The DistilBERT model is employed in experimental evaluations on the IMDb dataset, where 100 data points are selected for explanation. The comparison between GLIME-BINOMIAL and LIME

---

[3] `https://github.com/Horizon2333/imagenet-autoencoder`

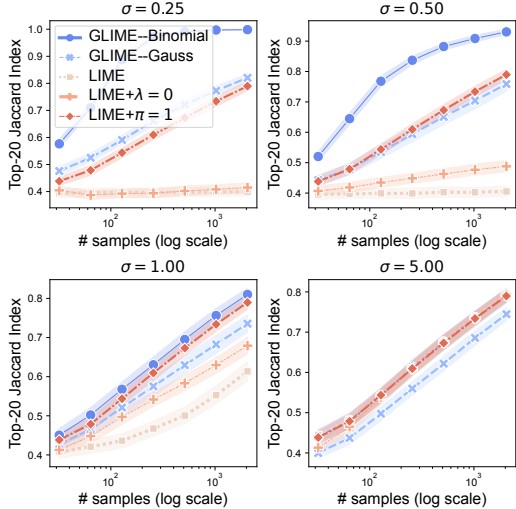

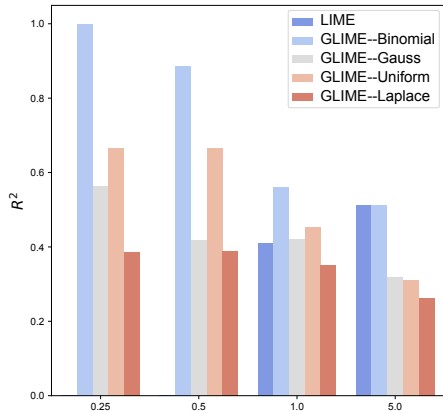

(a) **Stability across various methods (tiny Swin-Transformer).** The reported metric is the Top-20 Jaccard index. LIME+$\lambda = 0$ and LIME+$\pi = 1$ represent LIME without regularization and without weighting, respectively. LIME exhibits instability, particularly when $\sigma$ is small, whereas GLIME demonstrates enhanced stability across varying $\sigma$ values. Notably, in the absence of weighting or regularization, LIME's stability significantly improves when $\sigma$ is small. Conversely, the impact of regularization and weighting on LIME's stability is marginal when $\sigma$ is large.

(b) $R^2$ **comparison among LIME and GLIME variants with different sampling distributions (tiny Swin-Transformer).** For each image and method, 2048 samples are utilized to compute the explanation and the corresponding $R^2$. Notably, LIME yields nearly zero $R^2$ when $\sigma = 0.25$ and $0.5$, indicating an almost negligible explanation produced by LIME. In contrast, the $R^2$ values of LIME are consistently lower than those of GLIME, underscoring that GLIME enhances the local fidelity of LIME.

Figure 12: GLIME markedly enhances both stability and local fidelity compared to LIME across various values of $\sigma$.

Table 1: **Top-20 Jaccard Index of GLIME-BINOMIAL, GLIME-GAUSS, and ALIME.** GLIME-BINOMIAL and GLIME-GAUSS exhibit significantly higher stability than ALIME, particularly in scenarios with small $\sigma$ or limited samples.

| | # samples | 128 | 256 | 512 | 1024 |
|---|---|---|---|---|---|
| | GLIME-BINOMIAL | 0.952 | 0.981 | 0.993 | 0.998 |
| $\sigma = 0.25$ | GLIME-GAUSS | 0.872 | 0.885 | 0.898 | 0.911 |
| | ALIME | 0.618 | 0.691 | 0.750 | 0.803 |
| | GLIME-BINOMIAL | 0.596 | 0.688 | 0.739 | 0.772 |
| $\sigma = 0.5$ | GLIME-GAUSS | 0.875 | 0.891 | 0.904 | 0.912 |
| | ALIME | 0.525 | 0.588 | 0.641 | 0.688 |
| | GLIME-BINOMIAL | 0.533 | 0.602 | 0.676 | 0.725 |
| $\sigma = 1$ | GLIME-GAUSS | 0.883 | 0.894 | 0.908 | 0.915 |
| | ALIME | 0.519 | 0.567 | 0.615 | 0.660 |
| | GLIME-BINOMIAL | 0.493 | 0.545 | 0.605 | 0.661 |
| $\sigma = 5$ | GLIME-GAUSS | 0.865 | 0.883 | 0.898 | 0.910 |
| | ALIME | 0.489 | 0.539 | 0.589 | 0.640 |

is depicted in Figure 15 using the Jaccard Index. Our findings indicate that GLIME-BINOMIAL consistently exhibits higher stability than LIME across a range of $\sigma$ values and sample sizes. Notably, at smaller $\sigma$ values, GLIME-BINOMIAL demonstrates a substantial improvement in stability compared to LIME.

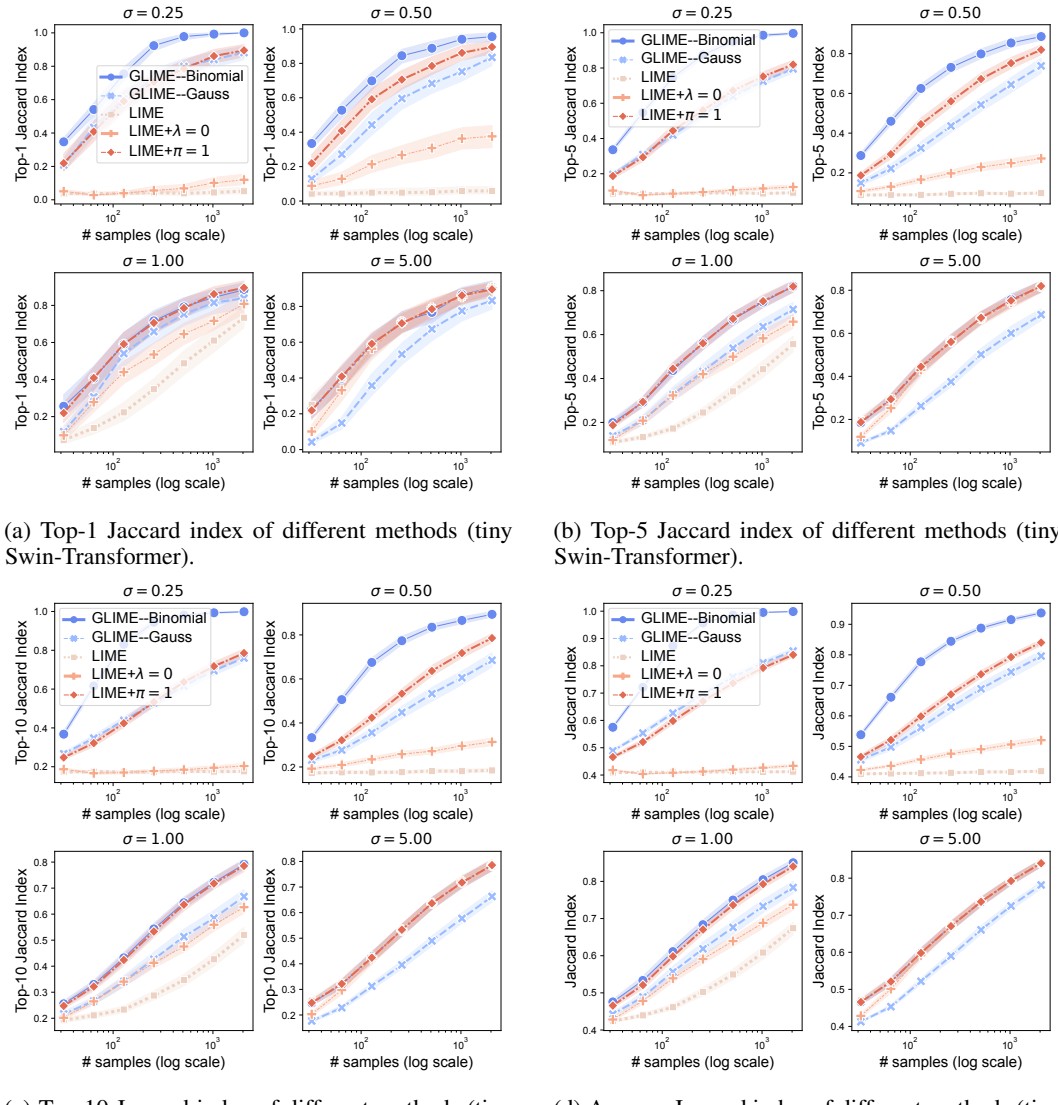

(a) Top-1 Jaccard index of different methods (tiny Swin-Transformer).

(b) Top-5 Jaccard index of different methods (tiny Swin-Transformer).

(c) Top-10 Jaccard index of different methods (tiny Swin-Transformer).

(d) Average Jaccard index of different methods (tiny Swin-Transformer).

Figure 13: Top-1, 5, 10, and average Jaccard indices are calculated for various methods, and the average Jaccard index represents the mean of top-1, $\cdots$, $d$ indices for the tiny Swin-Transformer.

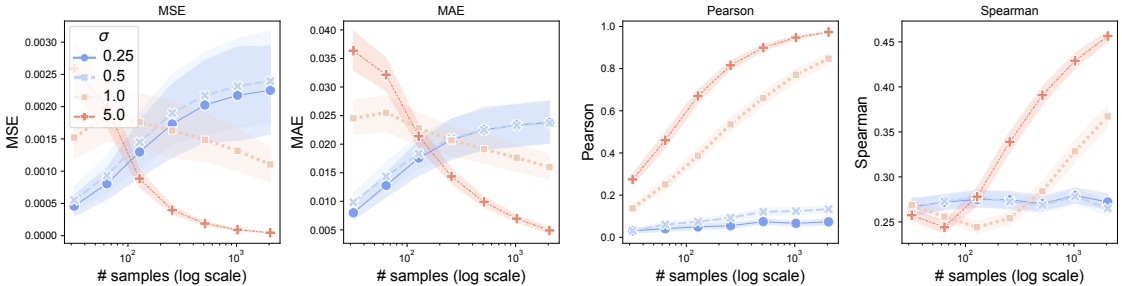

Figure 14: **Difference and correlation in LIME and GLIME-BINOMIAL explanations (tiny Swin-Transformer).** Mean Squared Error (MSE) and Mean Absolute Error (MAE) quantify the divergence between LIME and GLIME-BINOMIAL explanations. Pearson and Spearman correlation coefficients gauge the correlation. With an increasing sample size, the explanations tend to align more closely. Notably, the difference and correlation converge more rapidly with larger values of $\sigma$.

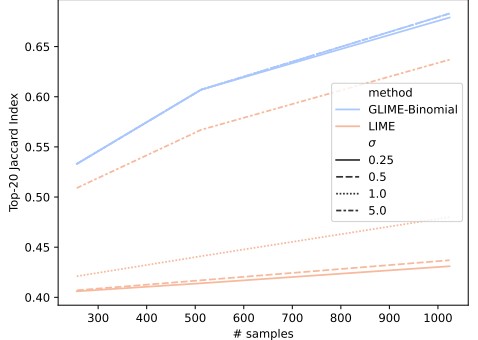
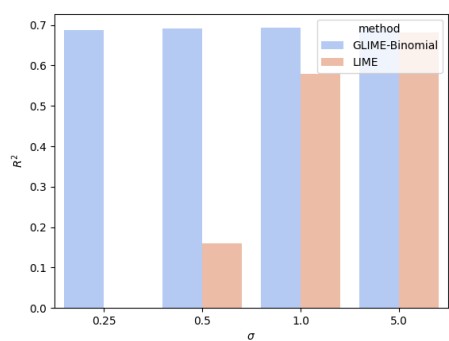

(a) **Stability comparison between GLIME-BINOMIAL and LIME.** The top-20 Jaccard index is reported, illustrating that GLIME displays superior stability compared to LIME across diverse values of $\sigma$. Notably, GLIME's stability remains consistently robust, showing limited sensitivity to changes in $\sigma$. In contrast, LIME exhibits increased stability as $\sigma$ values grow larger.

(b) **Comparison of $R^2$ between LIME and GLIME-BINOMIAL under various sampling distributions.** Using 2048 samples for explanation computation, $R^2$ values are computed for each image and method. It is noteworthy that when $\sigma = 0.25$, LIME exhibits nearly negligible $R^2$ values.

Figure 15: GLIME significantly enhances both stability and local fidelity compared to LIME across various $\sigma$ values.

## B    Proofs

### B.1    Equivalent GLIME formulation without $\pi(\cdot)$

By integrating the weighting function into the sampling distribution, the problem to be solved is

$$
\begin{aligned}
\mathbf{w}^{\text{GLIME}} &= \arg\min_{\mathbf{v}} \mathbb{E}_{\mathbf{z}' \sim \mathcal{P}}[\pi(\mathbf{z}')\ell(f(\mathbf{z}), g(\mathbf{z}'))] + \lambda R(\mathbf{v}) \\
&= \arg\min_{\mathbf{v}} \int_{\mathbb{R}^d} \pi(\mathbf{z}')\ell(f(\mathbf{z}), g(\mathbf{z}'))\mathcal{P}(\mathbf{z})\mathrm{d}\mathbf{z} + \lambda R(\mathbf{v}) \\
&= \frac{\arg\min_{\mathbf{v}} \int_{\mathbb{R}^d} \ell(f(\mathbf{z}), g(\mathbf{z}'))\pi(\mathbf{z}')\mathcal{P}(\mathbf{z})\mathrm{d}\mathbf{z} + \lambda R(\mathbf{v})}{\int_{\mathbb{R}^d} \pi(\mathbf{u}')\mathcal{P}(\mathbf{u})\mathrm{d}\mathbf{u}} \\
&= \arg\min_{\mathbf{v}} \frac{\int_{\mathbb{R}^d} \ell(f(\mathbf{z}), g(\mathbf{z}'))\pi(\mathbf{z}')\mathcal{P}(\mathbf{z})\mathrm{d}\mathbf{z}}{\int_{\mathbb{R}^d} \pi(\mathbf{u}')\mathcal{P}(\mathbf{u})\mathrm{d}\mathbf{u}} + \frac{\lambda R(\mathbf{v})}{\int_{\mathbb{R}^d} \pi(\mathbf{u}')\mathcal{P}(\mathbf{u})\mathrm{d}\mathbf{u}} \\
&= \arg\min_{\mathbf{v}} \int_{\mathbb{R}^d} \ell(f(\mathbf{z}), g(\mathbf{z}'))\widetilde{\mathcal{P}}(\mathbf{z})\mathrm{d}\mathbf{z} + \frac{\lambda}{Z}R(\mathbf{v}) \quad \widetilde{\mathcal{P}}(\mathbf{z}) = \frac{\pi(\mathbf{z}')\mathcal{P}(\mathbf{z})}{Z}, Z = \int_{\mathbb{R}^d} \pi(\mathbf{u}')\mathcal{P}(\mathbf{u})\mathrm{d}\mathbf{u} \\
&= \arg\min_{\mathbf{v}} \mathbb{E}_{\mathbf{z}' \sim \widetilde{\mathcal{P}}}[\ell(f(\mathbf{z}), g(\mathbf{z}'))] + \frac{\lambda}{Z}R(\mathbf{v})
\end{aligned}
$$

### B.2    Equivalence between LIME and GLIME-BINOMIAL

For LIME, $\mathcal{P} = \text{Uni}(\{0,1\}^d)$ and thus $\mathcal{P}(\mathbf{z}', \|\mathbf{z}'\|_0 = k) = \frac{1}{2^d}, k = 0, 1, \cdots, d$, so that

$$
Z = \int_{\mathbb{R}^d} \pi(\mathbf{u}')\mathcal{P}(\mathbf{u})\mathrm{d}\mathbf{u} = \sum_{k=0}^{d} e^{(k-d)/\sigma^2} \frac{\binom{d}{k}}{2^d} = \frac{e^{-d/\sigma^2}}{2^d}(1 + e^{1/\sigma^2})^d
$$

Thus, we have

$$
\widetilde{\mathcal{P}}(\mathbf{z}) = \frac{\pi(\mathbf{z}')\mathcal{P}(\mathbf{z})}{Z} = \frac{e^{(k-d)/\sigma^2} 2^{-d}}{Z} = \frac{e^{k/\sigma^2}}{(1 + e^{1/\sigma^2})^d}
$$

Therefore, GLIME-BINOMIAL is equivalent to LIME.

## B.3 LIME requires many samples to accurately estimate the expectation term in Equation 1.

In Figure 2, it is evident that a lot of samples generated by LIME possess considerably small weights. Consequently, the sample estimation of the expectation in Equation 1 tends to be much smaller than the true expectation with high probability. In such instances, the regularization term would have a dominant influence on the overall objective.

Consider specific parameters, such as $\sigma = 0.25$, $n = 1000$, $d = 20$ (where $\sigma$ and $n$ are the default values in the original implementation of LIME). The probability of obtaining a sample $\mathbf{z}'$ with $\|\mathbf{z}'\|_0 = d - 1$ or $d$ is approximately $\frac{d}{2^d} + \frac{1}{2^d} = \frac{d+1}{2^d} \approx 2 \times 10^{-5}$. Let's consider a typical scenario where $|f(\mathbf{z})| \in [0, 1]$, and $(f(\mathbf{z}) - \mathbf{v}^\top \mathbf{z}')^2$ is approximately $0.1$ for most $\mathbf{z}, \mathbf{z}'$. In this case, $\mathbb{E}_{\mathbf{z}' \sim \text{Uni}(\{0,1\}^d)}[\pi(\mathbf{z}')(f(\mathbf{z}) - \mathbf{v}^\top \mathbf{z}')^2] \approx 0.1 \cdot \sum_{k=0}^{d} \frac{e^{(k-d)/\sigma^2}}{2^d} \approx 10^{-7}$. However, if we lack samples $\mathbf{z}'$ with $\|\mathbf{z}'\|_0 = d - 1$ or $d$, then all samples $\mathbf{z}'_i$ with $\|\mathbf{z}'_i\|_0 \leq d - 2$ have weights $\pi(\mathbf{z}'_i) \leq \exp(-\frac{2}{\sigma^2}) \approx 1.26 \times 10^{-14}$. This leads to the sample average $\frac{1}{n} \sum_{i=1}^{n} \pi(\mathbf{z}'_i)(f(\mathbf{z}_i) - \mathbf{v}^\top \mathbf{z}'_i)^2 \leq 1.26 \times 10^{-15} \ll 10^{-7}$. The huge difference between the magnitude of the expectation term in Equation 1 and the sample average of this expectation indicates that the sample average is not an accurate estimation of $\mathbb{E}_{\mathbf{z}' \sim \text{Uni}(\{0,1\}^d)}[\pi(\mathbf{z}')(f(\mathbf{z}) - \mathbf{v}^\top \mathbf{z}')^2]$ (if we do not get enough samples). Additionally, under these circumstances, the regularization term is likely to dominate the sample average term, leading to an underestimation of the intended value of $\mathbf{v}$. In conclusion, the original sampling method for LIME, even with extensively used default parameters, is not anticipated to yield meaningful explanations.

## B.4 Proof of Theorem 4.1

**Theorem B.1.** *Suppose samples $\{\mathbf{z}'_i\}_{i=1}^{n} \sim Uni(\{0, 1\}^d)$ are used to compute LIME explanation. For any $\epsilon > 0, \delta \in (0, 1)$, if $n = \Omega(\epsilon^{-2} d^5 2^{4d} e^{4/\sigma^2} \log(4d/\delta))$, $\lambda \leq n$, we have $\mathbb{P}(\|\hat{\mathbf{w}}^{LIME} - \mathbf{w}^{LIME}\|_2 < \epsilon) \geq 1 - \delta$. $\mathbf{w}^{LIME} = \lim_{n \to \infty} \hat{\mathbf{w}}^{LIME}$.*

*Proof.* To compute the LIME explanation with $n$ samples, the following optimization problem is solved:

$$\hat{\mathbf{w}}^{\text{LIME}} = \arg\min_{\mathbf{v}} \frac{1}{n} \sum_{i=1}^{n} \pi(\mathbf{z}'_i)(f(\mathbf{z}_i) - \mathbf{v}^\top \mathbf{z}'_i)^2 + \frac{\lambda}{n} \|\mathbf{v}\|_2^2.$$

Let $L = \frac{1}{n} \sum_{i=1}^{n} \pi(\mathbf{z}'_i)(f(\mathbf{z}_i) - \mathbf{v}^\top \mathbf{z}'_i)^2 + \frac{\lambda}{n} \|\mathbf{v}\|_2^2$. Setting the gradient of $L$ with respect to $\mathbf{v}$ to zero, we obtain:

$$-2 \frac{1}{n} \pi(\mathbf{z}'_i)(f(\mathbf{z}_i) - \mathbf{v}^\top \mathbf{z}'_i)\mathbf{z}'_i + \frac{2}{n} \lambda \mathbf{v} = 0,$$

which leads to:

$$\hat{\mathbf{w}}^{\text{LIME}} = \left( \frac{1}{n} \sum_{i=1}^{n} \pi(\mathbf{z}'_i)\mathbf{z}'_i(\mathbf{z}'_i)^\top + \frac{\lambda}{n} \right)^{-1} \left( \frac{1}{n} \sum_{i=1}^{n} \pi(\mathbf{z}'_i)\mathbf{z}'_i f(\mathbf{z}_i) \right).$$

Denote $\mathbf{\Sigma}_n = \frac{1}{n} \sum_{i=1}^{n} \pi(\mathbf{z}'_i)\mathbf{z}'_i(\mathbf{z}'_i)^\top + \frac{\lambda}{n}$, $\mathbf{\Gamma}_n = \frac{1}{n} \sum_{i=1}^{n} \pi(\mathbf{z}'_i)\mathbf{z}'_i f(\mathbf{z}_i)$, $\mathbf{\Sigma} = \lim_{n \to \infty} \mathbf{\Sigma}_n$, and $\mathbf{\Gamma} = \lim_{n \to \infty} \mathbf{\Gamma}_n$. Then, we have:

$$\hat{\mathbf{w}}^{\text{LIME}} = \mathbf{\Sigma}_n^{-1} \mathbf{\Gamma}_n, \quad \mathbf{w}^{\text{LIME}} = \mathbf{\Sigma}^{-1} \mathbf{\Gamma}.$$

To prove the concentration of $\hat{\mathbf{w}}^{\text{LIME}}$, we follow the proofs in [8]: (1) First, we prove the concentration of $\mathbf{\Sigma}_n$; (2) Then, we bound $\|\mathbf{\Sigma}^{-1}\|_F^2$; (3) Next, we prove the concentration of $\mathbf{\Gamma}_n$; (4) Finally, we use the following inequality:

$$\|\mathbf{\Sigma}_n^{-1} \mathbf{\Gamma}_n - \mathbf{\Sigma}^{-1} \mathbf{\Gamma}\| \leq 2\|\mathbf{\Sigma}^{-1}\|_F \|\mathbf{\Gamma}_n - \mathbf{\Gamma}\|_2 + 2\|\mathbf{\Sigma}^{-1}\|_F^2 \|\mathbf{\Gamma}\| \|\mathbf{\Sigma}_n - \mathbf{\Sigma}\|,$$

when $\|\mathbf{\Sigma}^{-1}(\mathbf{\Sigma}_n - \mathbf{\Sigma})\| \leq 0.32$ [8].

Before establishing concentration results, we first derive the expression for $\mathbf{\Sigma}$.

**Expression of $\mathbf{\Sigma}$.**

$$\mathbf{\Sigma}_n = \begin{bmatrix} \frac{1}{n} \sum_i \pi(\mathbf{z}'_i)(\mathbf{z}'_{i1})^2 + \frac{\lambda}{n} & \frac{1}{n} \sum_i \pi(\mathbf{z}'_i)\mathbf{z}'_{i1}\mathbf{z}'_{i2} & \cdots & \frac{1}{n} \sum_i \pi(\mathbf{z}'_i)\mathbf{z}'_{i,1}\mathbf{z}'_{id} \\ \frac{1}{n} \sum_i \pi(\mathbf{z}'_i)\mathbf{z}'_{i1}\mathbf{z}'_{i2} & \frac{1}{n} \sum_i \pi(\mathbf{z}'_i)(\mathbf{z}'_{i2})^2 + \frac{\lambda}{n} & \cdots & \frac{1}{n} \sum_i \pi(\mathbf{z}'_i)\mathbf{z}'_{i2}\mathbf{z}'_{id} \\ \vdots & \vdots & \ddots & \vdots \\ \frac{1}{n} \sum_i \pi(\mathbf{z}'_i)\mathbf{z}'_{i1}\mathbf{z}'_{id} & \frac{1}{n} \sum_i \pi(\mathbf{z}'_i)\mathbf{z}'_{i2}\mathbf{z}'_{id} & \cdots & \frac{1}{n} \sum_i \pi(\mathbf{z}'_i)(\mathbf{z}'_{id})^2 + \frac{\lambda}{n} \end{bmatrix}$$

By taking $n \to \infty$, we have

$$\boldsymbol{\Sigma}_n \to \boldsymbol{\Sigma} = (\alpha_1 - \alpha_2)\mathbf{I} + \alpha_2 \mathbf{1}\mathbf{1}^\top$$

where

$$
\begin{aligned}
\alpha_1 =& \mathbb{E}_{\mathbf{z}' \sim \text{Uni}(\{0,1\}^d)}[\pi(\mathbf{z}')z_i'] = \mathbb{E}_{\mathbf{z}' \sim \text{Uni}(\{0,1\}^d)}[\pi(\mathbf{z}')(z_i')^2] \\
=& \sum_{k=0}^{d} e^{(k-d)/\sigma^2} \mathbb{P}(z_i' = 1 | \|\mathbf{z}'\|_0 = k)\mathbb{P}(\|\mathbf{z}'\|_0 = k) \\
=& \sum_{k=0}^{d} e^{(k-d)/\sigma^2} \frac{k}{d} \frac{\binom{d}{k}}{2^d} \\
=& \sum_{k=0}^{d} e^{(k-d)/\sigma^2} \frac{\binom{d-1}{k-1}}{2^d} \\
=& \sum_{k=0}^{d} e^{(k-1)/\sigma^2} e^{(1-d)/\sigma^2} \frac{\binom{d-1}{k-1}}{2^d} \\
=& e^{(1-d)/\sigma^2} \frac{(1 + e^{\frac{1}{\sigma^2}})^{d-1}}{2^d} = \frac{(1 + e^{-\frac{1}{\sigma^2}})^{d-1}}{2^d} \\
\alpha_2 =& \mathbb{E}_{\mathbf{z}' \sim \text{Uni}(\{0,1\}^d)}[\pi(\mathbf{z}')z_i'z_j'] \\
=& \frac{1}{Z} \sum_{k=0}^{d} e^{(k-d)/\sigma^2} \mathbb{P}(z_i' = 1, z_j' = 1 | \|\mathbf{z}'\|_0 = k)\mathbb{P}(\|\mathbf{z}'\|_0 = k) \\
=& \sum_{k=0}^{d} e^{(k-d)/\sigma^2} \frac{k(k-1)}{d(d-1)} \frac{\binom{d}{k}}{2^d} \\
=& \sum_{k=0}^{d} e^{(k-d)/\sigma^2} \frac{\binom{d-2}{k-2}}{2^d} \\
=& \sum_{k=0}^{d} e^{(k-2)/\sigma^2} e^{(2-d)/\sigma^2} \frac{\binom{d-2}{k-2}}{2^d} \\
=& e^{(2-d)/\sigma^2} \frac{(1 + e^{\frac{1}{\sigma^2}})^{d-2}}{2^d} = \frac{(1 + e^{-\frac{1}{\sigma^2}})^{d-2}}{2^d}
\end{aligned}
$$

By Sherman-Morrison formula, we have

$$
\begin{aligned}
\boldsymbol{\Sigma}^{-1} &= ((\alpha_1 - \alpha_2)\mathbf{I} + \alpha_2 \mathbf{1}\mathbf{1}^\top)^{-1} = \frac{1}{\alpha_1 - \alpha_2}(\mathbf{I} + \frac{\alpha_2}{\alpha_1 - \alpha_2}\mathbf{1}\mathbf{1}^\top)^{-1} \\
&= \frac{1}{\alpha_1 - \alpha_2}(\mathbf{I} - \frac{\frac{\alpha_2}{\alpha_1 - \alpha_2}\mathbf{1}\mathbf{1}^\top}{1 + \frac{\alpha_2}{\alpha_1 - \alpha_2}d}) = (\beta_1 - \beta_2)\mathbf{I} + \beta_2 \mathbf{1}\mathbf{1}^\top
\end{aligned}
$$

where

$$\beta_1 = \frac{\alpha_1 + (d-2)\alpha_2}{(\alpha_1 - \alpha_2)(\alpha_1 + (d-1)\alpha_2)}, \quad \beta_2 = -\frac{\alpha_2}{(\alpha_1 - \alpha_2)(\alpha_1 + (d-1)\alpha_2)}$$

In the following, we aim to establish the concentration of $\hat{\mathbf{w}}^{\text{LIME}}$.

**Concentration of $\boldsymbol{\Sigma}_n$.** Considering $0 \leq \pi(\cdot) \leq 1$ and $\mathbf{z}_i \in \{0,1\}^d$, each element within $\boldsymbol{\Sigma}_n$ resides within the interval of $[0, 2]$. Moreover, as

$$\frac{1}{2^d} \leq \alpha_1 = \frac{(1 + e^{-\frac{1}{\sigma^2}})^{d-1}}{2^d} \leq \frac{2^{d-1}}{2^d} = \frac{1}{2}$$

$$\frac{1}{2^d} \leq \alpha_2 = \frac{(1 + e^{-\frac{1}{\sigma^2}})^{d-2}}{2^d} \leq \frac{2^{d-2}}{2^d} = \frac{1}{4}$$

$$\frac{e^{-1/\sigma^2}}{2^d} \leq \alpha_1 - \alpha_2 = e^{-\frac{1}{\sigma^2}} \frac{(1 + e^{-\frac{1}{\sigma^2}})^{d-2}}{2^d} \leq \frac{1}{4}$$

The elements within $\boldsymbol{\Sigma}$ are within the range of $[0, \frac{1}{4}]$. Consequently, the elements in $\boldsymbol{\Sigma}_n - \boldsymbol{\Sigma}$ fall within the range of $[-\frac{1}{4}, 2]$.

Referring to the matrix Hoeffding's inequality [31], it holds true that for all $t > 0$,

$$\mathbb{P}(\|\boldsymbol{\Sigma}_n - \boldsymbol{\Sigma}\|_2 \geq t) \leq 2d \exp(-\frac{nt^2}{32d^2})$$

**Bounding $\|\boldsymbol{\Sigma}^{-1}\|_F^2$.**

$$\|\boldsymbol{\Sigma}^{-1}\|_F^2 = d\beta_1^2 + (d^2 - d)\beta_2^2$$

Because

$$\frac{d}{2^d} \leq \alpha_1 + (d-1)\alpha_2 \leq \frac{2 + (d-1)}{4} = \frac{d+1}{4}$$

$$\frac{d-1}{2^d} \leq \alpha_1 + (d-2)\alpha_2 \leq \frac{2 + (d-2)}{4} = \frac{d}{4}$$

we have

$$|\beta_1| = \left| \frac{\alpha_1 + (d-2)\alpha_2}{(\alpha_1 - \alpha_2)(\alpha_1 + (d-1)\alpha_2)} \right| \leq \left| \frac{1}{\alpha_1 - \alpha_2} \right| \leq 2^d e^{1/\sigma^2}, \beta_1^2 \leq 2^{2d} e^{2/\sigma^2}$$

$$|\beta_2| = \left| -\frac{\alpha_2}{(\alpha_1 - \alpha_2)(\alpha_1 + (d-1)\alpha_2)} \right| = \left| e^{1/\sigma^2} \frac{1}{(\alpha_1 + (d-1)\alpha_2)} \right| \leq d^{-1} 2^d e^{1/\sigma^2},$$

$$\beta_2^2 \leq d^{-2} 2^{2d} e^{2/\sigma^2}$$

so that

$$\|\boldsymbol{\Sigma}^{-1}\|_F^2 = d\beta_1^2 + (d^2 - d)\beta_2^2 \leq d2^{2d} e^{2/\sigma^2} + (d^2 - d)d^{-2} 2^{2d} e^{2/\sigma^2} \leq 2d2^{2d} e^{2/\sigma^2}$$

**Concentration of $\boldsymbol{\Gamma}_n$.** With $|f| \leq 1$, all elements within both $\boldsymbol{\Gamma}_n$ and $\boldsymbol{\Gamma}$ exist within the range of $[0, 1]$. According to matrix Hoeffding's inequality [31], for all $t > 0$,

$$\mathbb{P}(\|\boldsymbol{\Gamma}_n - \boldsymbol{\Gamma}\| \geq t) \leq 2d \exp\left(-\frac{nt^2}{8d}\right)$$

**Concentration of $\hat{\mathbf{w}}^{\text{LIME}}$.** When $\|\boldsymbol{\Sigma}^{-1}(\boldsymbol{\Sigma}_n - \boldsymbol{\Sigma})\| \leq 0.32$ [8], we have

$$\|\boldsymbol{\Sigma}_n^{-1}\boldsymbol{\Gamma}_n - \boldsymbol{\Sigma}^{-1}\boldsymbol{\Gamma}\| \leq 2\|\boldsymbol{\Sigma}^{-1}\|_F \|\boldsymbol{\Gamma}_n - \boldsymbol{\Gamma}\|_2 + 2\|\boldsymbol{\Sigma}^{-1}\|_F^2 \|\boldsymbol{\Gamma}\| \|\boldsymbol{\Sigma}_n - \boldsymbol{\Sigma}\|$$

Given that

$$\|\boldsymbol{\Sigma}^{-1}(\boldsymbol{\Sigma}_n - \boldsymbol{\Sigma})\| \leq \|\boldsymbol{\Sigma}^{-1}\| \|\boldsymbol{\Sigma}_n - \boldsymbol{\Sigma}\| \leq 2^{1/2} d^{1/2} 2^d e^{1/\sigma^2} \|\boldsymbol{\Sigma}_n - \boldsymbol{\Sigma}\|$$

Exploiting the concentration of $\boldsymbol{\Sigma}_n$, where $n \geq n_1 = 2^7 d^3 2^{2d} e^{2/\sigma^2} \log(4d/\delta)$ and $t = t_1 = 5^{-2} 2^{2.5} d^{-0.5} 2^{-d} e^{-1/\sigma^2}$, we have

$$\mathbb{P}(\|\boldsymbol{\Sigma}_n - \boldsymbol{\Sigma}\|_2 \geq t) \leq 2d \exp\left(-\frac{nt^2}{32d^2}\right) \leq 2d \exp\left(-\frac{nt^2}{32d^2}\right) \leq \frac{\delta}{2}$$

Therefore, with a probability of at least $1 - \frac{\delta}{2}$, we have

$$\|\mathbf{\Sigma}^{-1}(\mathbf{\Sigma}_n - \mathbf{\Sigma})\| \leq \|\mathbf{\Sigma}^{-1}\|\|\mathbf{\Sigma}_n - \mathbf{\Sigma}\| \leq 2^{1/2}d^{1/2}2^d e^{1/\sigma^2}\|\mathbf{\Sigma}_n - \mathbf{\Sigma}\| \leq 0.32$$

For $n \geq n_2 = 2^8\epsilon^{-2}d^2 2^{2d}e^{2/\sigma^2}\log(4d/\delta)$ and $t_2 = 2^{-2.5}d^{-0.5}2^{-d}e^{-1/\sigma^2}\epsilon$, the following concentration inequality holds:

$$\mathbb{P}(\|\mathbf{\Gamma}_n - \mathbf{\Gamma}\| \geq t_2) \leq 2d\exp\left(-\frac{n_2 t_2^2}{8d}\right) \leq \frac{\delta}{2}$$

In this context, with a probability of at least $1 - \frac{\delta}{2}$, we have

$$\|\mathbf{\Sigma}^{-1}\|\|\mathbf{\Gamma}_n - \mathbf{\Gamma}\| \leq \frac{\epsilon}{4}$$

Considering $\|\mathbf{\Gamma}\| \leq \sqrt{d}$, we select $n \geq n_3 = 2^9\epsilon^{-2}d^5 2^{4d}e^{4/\sigma^2}\log(4d/\delta)$ and $t_3 = 2^{-3}d^{-1.5}2^{-2d}e^{-2/\sigma^2}\epsilon$, leading to

$$\mathbb{P}(\|\mathbf{\Sigma}_n - \mathbf{\Sigma}\|_2 \geq t_3) \leq 2d\exp\left(-\frac{n_3 t_3^2}{32d^2}\right) \leq 2d\exp\left(-\frac{n_3 t_3^2}{32d^2}\right) \leq \frac{\delta}{2}$$

With a probability at least $1 - \delta/2$, we have

$$\|\mathbf{\Sigma}^{-1}\|^2\|\mathbf{\Gamma}\|\|\mathbf{\Sigma}_n - \mathbf{\Sigma}\| \leq \frac{\epsilon}{4}$$

In summary, we choose $n \geq \max\{n_1, n_2, n_3\}$, and then for all $\epsilon > 0, \delta \in (0,1)$

$$\mathbb{P}(\|\mathbf{\Sigma}_n^{-1}\mathbf{\Gamma}_n - \mathbf{\Sigma}^{-1}\mathbf{\Gamma}\| \geq \epsilon) \leq \delta$$

$\square$

### B.5 Proof of Theorem 4.2 and Corollary 4.3

**Theorem B.2.** *Suppose $\mathbf{z}' \sim \mathcal{P}$ such that the largest eigenvalue of $\mathbf{z}'(\mathbf{z}')^\top$ is bounded by $R$ and $\mathbb{E}[\mathbf{z}'(\mathbf{z}')^\top] = (\alpha_1 - \alpha_2)\mathbf{I} + \alpha_2\mathbf{1}\mathbf{1}^\top, \|Var(\mathbf{z}'(\mathbf{z}')^\top)\|_2 \leq \nu^2, |(\mathbf{z}'f(\mathbf{z}))_i| \leq M$ for some $M > 0$. $\{\mathbf{z}_i'\}_{i=1}^n$ are i.i.d. samples from $\mathcal{P}$ and are used to compute GLIME explanation $\hat{\mathbf{w}}^{\text{GLIME}}$. For any $\epsilon > 0, \delta \in (0,1)$, if $n = \Omega(\epsilon^{-2}M^2\nu^2 d^3\gamma^4\log(4d/\delta))$ where $\gamma^2 = d\beta_1^2 + (d^2 - d)\beta_2^2, \beta_1 = (\alpha_1 + (d-2)\alpha_2)/\beta_0, \beta_2 = -\alpha_2/\beta_0, \beta_0 = (\alpha_1 - \alpha_2)(\alpha_1 + (d-1)\alpha_2))$, we have $\mathbb{P}(\|\hat{\mathbf{w}}^{\text{GLIME}} - \mathbf{w}^{\text{GLIME}}\|_2 < \epsilon) \geq 1 - \delta$. $\mathbf{w}^{\text{GLIME}} = \lim_{n\to\infty}\hat{\mathbf{w}}^{\text{GLIME}}$.*

*Proof.* The proof closely resembles that of Theorem 4.1. Employing the same derivation, we deduce that:

$$\mathbf{\Sigma} = (\alpha_1 + \lambda - \alpha_2)\mathbf{I} + \alpha_2\mathbf{1}\mathbf{1}^\top, \quad \mathbf{\Sigma}^{-1} = (\beta_1 - \beta_2)\mathbf{I} + \beta_2\mathbf{1}\mathbf{1}^\top$$

where

$$\beta_1 = \frac{\alpha_1 + \lambda + (d-2)\alpha_2}{(\alpha_1 + \lambda - \alpha_2)(\alpha_1 + \lambda + (d-1)\alpha_2)}, \quad \beta_2 = -\frac{\alpha_2}{(\alpha_1 + \lambda - \alpha_2)(\alpha_1 + \lambda + (d-1)\alpha_2)}$$

Given that $\lambda_{\max}(z'(z')^\top) \leq R$ and $\|Var(\mathbf{z}'(\mathbf{z}')^\top)\|_2 \leq \nu^2$, according to the matrix Hoeffding's inequality [31], for all $t > 0$:

$$\mathbb{P}(\|\mathbf{\Sigma}_n - \mathbf{\Sigma}\|_2 \geq t) \leq 2d\exp\left(-\frac{nt^2}{8\nu^2}\right)$$

Applying Hoeffding's inequality coordinate-wise, we obtain:

$$\mathbb{P}(\|\mathbf{\Gamma}_n - \mathbf{\Gamma}\| \geq t) \leq 2d \exp\left(-\frac{nt^2}{8M^2d^2}\right)$$

Additionally,

$$\|\mathbf{\Sigma}^{-1}\|_F^2 = d\beta_1^2 + (d^2 - d)\beta_2^2 = \gamma^2$$

By selecting $n \geq n_1 = 2^5 \gamma^2 \nu^2 \log(4d/\delta)$ and $t_1 = 2^3 5^{-2} \gamma^{-1}$, we obtain

$$\mathbb{P}(\|\mathbf{\Sigma}_n - \mathbf{\Sigma}\|_2 \geq t_1) \leq 2d \exp\left(-\frac{n_1 t_1^2}{8\nu^2}\right) \leq \frac{\delta}{2}$$

with a probability of at least $1 - \delta/2$.

$$\|\mathbf{\Sigma}^{-1}(\mathbf{\Sigma}_n - \mathbf{\Sigma})\| \leq \|\mathbf{\Sigma}^{-1}\| \cdot \|\mathbf{\Sigma}_n - \mathbf{\Sigma}\| \leq \gamma t_1 = 0.32$$

Letting $n \geq n_2 = 2^5 \epsilon^{-2} M^2 d^2 \gamma^2 \log(4d/\delta)$ and $t_2 = 2^{-2} \epsilon \gamma^{-1}$, we have

$$\mathbb{P}(\|\mathbf{\Gamma}_n - \mathbf{\Gamma}\| \geq t_2) \leq 2d \exp\left(-\frac{n_2 t_2^2}{8M^2d^2}\right) \leq \frac{\delta}{2}$$

with a probability of at least $1 - \delta/2$.

$$\|\mathbf{\Sigma}\|\|\mathbf{\Gamma}_n - \mathbf{\Gamma}\| \leq \gamma t_2 \leq \frac{\epsilon}{4}$$

As $\|\mathbf{\Gamma}\| \leq M$, by choosing $n \geq n_3 = 2^5 \epsilon^{-2} M^2 \nu^2 d\gamma^4 \log(4d/\delta)$ and $t_3 = 2^{-2} \epsilon M^{-1} d^{-0.5} \gamma^{-2}$, we have

$$\mathbb{P}(\|\mathbf{\Sigma}_n - \mathbf{\Sigma}\|_2 \geq t_3) \leq 2d \exp\left(-\frac{n_3 t_3^2}{2\nu^2}\right) \leq \frac{\delta}{2}$$

and with a probability of at least $1 - \delta/2$,

$$\|\mathbf{\Sigma}^{-1}\|^2 \|\mathbf{\Gamma}\|\|\mathbf{\Sigma}_n - \mathbf{\Sigma}\| \leq \gamma^2 M d^{0.5} t_3 = \frac{\epsilon}{4}$$

Therefore, by choosing $n = \max\{n_1, n_2, n_3\}$, we have

$$\mathbb{P}(\|\mathbf{\Sigma}_n^{-1}\mathbf{\Gamma}_n - \mathbf{\Sigma}^{-1}\mathbf{\Gamma}\| \geq \epsilon) \leq \delta$$

$\square$

**Corollary B.3.** *Suppose $\{\mathbf{z}_i'\}_{i=1}^n$ are i.i.d. samples from $\mathbb{P}(\mathbf{z}', \|\mathbf{z}'\|_0 = k) = e^{k/\sigma^2}/(1+e^{1/\sigma^2})^d, k = 1, \ldots, d$ are used to compute* GLIME-BINOMIAL *explanation. For any $\epsilon > 0, \delta \in (0, 1)$, if $n = \Omega(\epsilon^{-2} d^5 e^{4/\sigma^2} \log(4d/\delta))$, we have $\mathbb{P}(\|\hat{\mathbf{w}}^{Binomial} - \mathbf{w}^{Binomial}\|_2 < \epsilon) \geq 1 - \delta$. $\mathbf{w}^{Binomial} = \lim_{n\to\infty} \hat{\mathbf{w}}^{Binomial}$.*

*Proof.* For GLIME-BINOMIAL, each coordinate of $\mathbf{z}'(\mathbf{z}')^\top$ follows a Bernoulli distribution, ensuring the bounded variance of both $\mathbf{z}'(\mathbf{z}')^\top$ and $(\mathbf{z}' f(\mathbf{z}'))_i$. Additionally, we have

$$\|\mathbf{\Gamma}\| \leq \sqrt{d},$$

$$\alpha_1 = \mathbb{E}[(z_i^2)'] = \mathbb{E}[z_i'] = \frac{e^{1/\sigma^2}}{1 + e^{1/\sigma^2}}$$

$$= \sum_{k=0}^{d} \mathbb{P}(z_i' = 1 | \|\mathbf{z}'\|_0 = k) \mathbb{P}(\|\mathbf{z}'\|_0 = k)$$

$$= \sum_{k=0}^{d} \frac{k}{d} \frac{\binom{d}{k} e^{k/\sigma^2}}{(1 + e^{1/\sigma^2})^d}$$

$$= \sum_{k=0}^{d} \frac{\binom{d-1}{k-1} e^{k/\sigma^2}}{(1 + e^{1/\sigma^2})^d}$$

$$= \frac{(1 + e^{1/\sigma^2})^{d-1}}{(1 + e^{1/\sigma^2})^d} e^{1/\sigma^2} = \frac{e^{1/\sigma^2}}{1 + e^{1/\sigma^2}}$$

$$\alpha_2 = \mathbb{E}[z_i' z_j'] = \frac{e^{1/\sigma^2}}{1 + e^{1/\sigma^2}}$$

$$= \sum_{k=0}^{d} \mathbb{P}(z_i' = 1, z_j' = 1 | \|\mathbf{z}'\|_0 = k) \mathbb{P}(\|\mathbf{z}'\|_0 = k)$$

$$= \sum_{k=0}^{d} \frac{k(k-1)}{d(d-1)} \frac{\binom{d}{k} e^{k/\sigma^2}}{(1 + e^{1/\sigma^2})^d}$$

$$= \sum_{k=0}^{d} \frac{\binom{d-2}{k-2} e^{k/\sigma^2}}{(1 + e^{1/\sigma^2})^d}$$

$$= \frac{(1 + e^{1/\sigma^2})^{d-2}}{(1 + e^{1/\sigma^2})^d} e^{2/\sigma^2} = \frac{e^{2/\sigma^2}}{(1 + e^{1/\sigma^2})^2} = \alpha_1^2$$

$$|\beta_1|^2 = \left| \frac{\alpha_1 + \lambda + (d-2)\alpha_2}{(\alpha_1 + \lambda - \alpha_2)(\alpha_1 + \lambda + (d-1)\alpha_2)} \right|^2$$

$$\leq \left| \frac{1}{\alpha_1 + \lambda - \alpha_2} \right| \leq \frac{1}{|\alpha_1 - \alpha_2|} = e^{-1/\sigma^2}(1 + e^{1/\sigma^2})^2 \leq 4 e^{1/\sigma^2}$$

$$|\beta_2|^2 = \left| -\frac{\alpha_2}{(\alpha_1 + \lambda - \alpha_2)(\alpha_1 + \lambda + (d-1)\alpha_2)} \right|^2$$

$$\leq \frac{\alpha_2^2}{(\alpha_1 - \alpha_2)(\alpha_1 + \lambda + (d-1)\alpha_2)^2}$$

$$= \frac{\alpha_1 \alpha_2}{(1 - \alpha_1)(\alpha_1 + (d-1)\alpha_2)^2}$$

$$\leq \frac{\alpha_1 \alpha_2}{(1 - \alpha_1)((d-1)\alpha_2)^2} = \frac{e^{-1/\sigma^2}(1 + e^{1/\sigma^2})^2}{(d-1)^2} \leq \frac{2^2 e^{1/\sigma^2}}{(d-1)^2}$$

Therefore,

$$d\beta_1^2 + (d^2 - d)\beta_2^2 \leq d e^{1/\sigma^2} + e^{1/\sigma^2} \frac{d}{d-1} \leq d e^{1/\sigma^2}$$

$\square$

## B.6 Formulation of SmoothGrad

**Proposition B.4.** *SmoothGrad is equivalent to* GLIME *formulation with* $\mathbf{z} = \mathbf{z}' + \mathbf{x}$ *where* $\mathbf{z}' \sim \mathcal{N}(\mathbf{0}, \sigma^2 \mathbf{I})$, $\ell(f(\mathbf{z}), g(\mathbf{z}')) = (f(\mathbf{z}) - g(\mathbf{z}'))^2$ *and* $\pi(\mathbf{z}) = 1, \Omega(\mathbf{v}) = 0$.

*The explanation returned by* GLIME *for* $f$ *at* $\mathbf{x}$ *with infinitely many samples under the above setting is*

$$\mathbf{w}^* = \frac{1}{\sigma^2} \mathbb{E}_{\mathbf{z}' \sim \mathcal{N}(\mathbf{0}, \sigma^2 \mathbf{I})}[\mathbf{z}' f(\mathbf{z}' + \mathbf{x})] = \mathbb{E}_{\mathbf{z}' \sim \mathcal{N}(\mathbf{0}, \sigma^2 \mathbf{I})}[\nabla f(\mathbf{x} + \mathbf{z}')]$$

*which is exactly SmoothGrad explanation. When* $\sigma \to 0$, $\mathbf{w}^* \to \nabla f(\mathbf{x} + \mathbf{z})|_{\mathbf{z}=\mathbf{0}}$.

*Proof.* To establish this proposition, we commence by deriving the expression for the GLIME explanation vector $\mathbf{w}^*$.

**Exact Expression of $\Sigma$:** For each $i = 1, \cdots, n$, let $\mathbf{z}'_i \sim \mathcal{N}(\mathbf{0}, \sigma^2 \mathbf{I})$. In this context,

$$\hat{\mathbf{\Sigma}}_n = \begin{bmatrix} \frac{1}{n} \sum_k (z_{k1}^2)' & \cdots & \frac{1}{n} \sum_k z'_{l1} z'_{kd} \\ \vdots & \ddots & \vdots \\ \frac{1}{n} \sum_k z'_{kd} z'_{k1} & \cdots & \frac{1}{n} \sum_k (z_{kd}^2)' \end{bmatrix}$$

This implies

$$\mathbf{\Sigma} = \mathbb{E}_{\mathbf{z}' \sim \mathcal{N}(\mathbf{0}, \sigma^2 \mathbf{I})}[\mathbf{z}'(\mathbf{z}')^\top] = \begin{bmatrix} \sigma^2 & \cdots & 0 \\ \vdots & \ddots & \vdots \\ 0 & \cdots & \sigma^2 \end{bmatrix}$$

$$\mathbf{\Sigma}^{-1} = \begin{bmatrix} \frac{1}{\sigma^2} & \cdots & 0 \\ \vdots & \ddots & \vdots \\ 0 & \cdots & \frac{1}{\sigma^2} \end{bmatrix}$$

Consequently, we obtain

$$\mathbf{w}^* = \mathbf{\Sigma}^{-1}\mathbf{\Gamma} = \frac{1}{\sigma^2} \mathbb{E}_{\mathbf{z}' \sim \mathcal{N}(\mathbf{0}, \sigma^2 \mathbf{I})}[\mathbf{z}' f(\mathbf{x} + \mathbf{z}')] = \mathbb{E}_{\mathbf{z}' \sim \mathcal{N}(\mathbf{0}, \sigma^2 \mathbf{I})}[\nabla f(\mathbf{x} + \mathbf{z}')]$$

The final equality is a direct consequence of Stein's lemma [17]. $\square$

