# OpenReview forum: "GLIME: General, Stable and Local LIME Explanation"
_NeurIPS.cc/2023/Conference — NeurIPS 2023 spotlight_

### Official Review · Reviewer_jFSL · 2023-06-26

**Soundness:** 2 fair
**Presentation:** 3 good
**Contribution:** 2 fair
**Rating:** 7
**Confidence:** 4

**Summary:**

This work is embedded in the research on model-agnostic explanations, i.e., to provide the user an understanding on the outputs of otherwise black-box prediction methods without knowing about the model's internals.  While LIME is a popular approach to solve this problem, prior work has demonstrated LIME to suffer from a strong dependence on the random seed, leading to instability and inconsistency. Since reliable, model-agnostic explanations will be a crucial tool for research and application alike to afford the use of otherwise black-box machine learning models, this paper is tackling an important issue considering LIME's popularity yet evident short-comings. GLIME is presented as a step towards more general, stable and consistent model-explanations. Due to the free choice of its sampling distribution and weights, it is shown how GLIME not only improves on LIME but generalizes over other methods.

**Strengths:**

1.The method GLIME is presented very clearly. Not only are text and equations supporting the reader in understanding the method well, but its motivation as successor of LIME in terms of stability and local fidelity are easy to follow and well justified by both the presented related work as well as this paper's own evaluation.

2. The unification of various model-explanation methods not only gives the reader an overview of how these methods relate to each other but shows well how GLIME is not only succeeding at stability and local fidelity but also a more general framework then LIME.

**Weaknesses:**

1. This work is strongly focused on comparing GLIME and its variants to LIME. While the relation of LIME and GLIME are made clear and well supported by the experiments, a more comprehensive overview on the field of explanation methods other than LIME could help the reader to better understand how GLIME fits into current research. Similarly, a discussion of GLIME's short-comings and an outline of Future Work would reinforce the contribution. Along the same line, a discussion on GLIME's quality as model-explainer and human-interpretability of the achieved results would greatly support the claims.

2. While the figures present the concepts and results of this paper quite well, they could benefit from some additional attention and polishing. For example, Fig 1a misses an explanation of the employed colormap. Fig. 1b shows GLIME-Gauss as blue dot in the legend but not the graphic itself. In Fig. 4a, the legend occupies important parts of the plot such that GLIME-Binomial and GLIME-Gauss curves are hard to see.

3. The use of inline-math can at times be overwhelming, e.g., in Theorem 4.2. While it is important to state all the relevant equations and relations, reverting to display- rather than inline-math for the key concepts might help the reader to better digest the underlying theory and assumptions.

**Questions:**

1. What are GLIME's short-comings and what are plans to improve on the method in the future?

2. Has it been evaluated if there is a difference in how human-interpretable GLIME's and LIME's explanations are?

**Limitations:**

Overall I would say it's a technically sound paper, but for a model-explanation paper I am missing the human-side a bit. My understanding is that the improvements the work shows are only meaningful if the consistent/stable explanations are also still good explanations, which I think is not really discussed here.

---

> ### Author Rebuttal · Authors · 2023-08-09
>
> We thank Reviewer jFSL for reviewing our paper and for the insightful comments. We hope our answers will address your concerns.
>
> > **Q1:** What are GLIME's short-comings and what are plans to improve on the method in the future?
>
> **A1:** From our perspective, the following may be promising for future research:
>
> * Parameter selection: It would be worthwhile to explore algorithms that can adaptively determine the optimal locality parameter $\sigma$ for each input $\mathbf{x}$, allowing for more flexible and context-specific explanations.
>
> * Feature correlation: Considering feature correlation in the explanation process could enhance the accuracy and interpretability of the results. Developing algorithms that incorporate feature correlation would be a promising avenue for future research.
>
> > **Q2:** Has it been evaluated if there is a difference in how human-interpretable GLIME's and LIME's explanations are? My understanding is that the improvements the work shows are only meaningful if the consistent/stable explanations are also still good explanations, which I think is not really discussed here.
>
> **A2:** We acknowledge the absence of a comparison between LIME and GLIME in terms of human interpretability in our study. We understand the significance of evaluating the interpretability of these methods and will carefully consider how to conduct such a comparison in future research. LIME, being one of the most widely utilized explanation methods, relies on linear model approximation to explain model behavior. Previous studies on LIME (e.g., ALIME [1], DLIME [2], S-LIME [3])  have primarily focused on addressing its limitations, particularly instability issues, rather than conducting human evaluation experiments. Instead, alternative evaluation metrics have been emphasized.
>
> Based on our current findings, we maintain the belief that GLIME outperforms LIME. A prominent concern with LIME is its high sensitivity to the choice of random seed. Figure 1(a) clearly demonstrates that different random seeds can yield entirely disparate explanation results. In contrast, GLIME consistently produces outputs irrespective of the seed, as indicated by the Jaccard Index. Explanations that are heavily influenced by random seeds may undermine their reliability and significance. Consequently, we argue that GLIME provides more meaningful explanations in comparison to LIME.
>
> Nonetheless, we recognize the necessity of conducting experiments to compare these methods in terms of human interpretability. We have plans to conduct relevant experiments in the future to address this aspect. Due to time constraints, we are currently unable to present experimental results, but we assure you that they will be included in the final version of our study.
>
> > **Q3:** A more comprehensive overview on the field of explanation methods other than LIME could help the reader to better understand how GLIME fits into current research.
>
> **A3:** Our main focus is to explore the limitations of LIME and how our proposed method, GLIME, addresses those limitations. Therefore, in the main body of the text, we primarily discuss LIME and similar explanation methods. There are many commonly used methods (e.g., SHAP, Anchor, SmoothGrad) that differ significantly from LIME, which we did not mention. However, we will provide a more detailed introduction to some commonly used explanation methods in the appendix, so that readers can have a clearer understanding of the differences between GLIME and those methods, as well as the applicability of GLIME.
>
> > **Q4:** Fig 1a misses an explanation of the employed colormap. Fig. 1b shows GLIME-Gauss as blue dot in the legend but not the graphic itself. In Fig. 4a, the legend occupies important parts of the plot such that GLIME-Binomial and GLIME-Gauss curves are hard to see. The use of inline-math can at times be overwhelming, e.g., in Theorem 4.2.
>
> **A4:** Thank you for bringing these issues to our attention. We appreciate your feedback and will make the necessary changes to address them. We will also conduct a thorough review and revision of other sections in accordance with your suggestions.
>
> [1] Sharath et al., ALIME: Autoencoder Based Approach for Local Interpretability
>
> [2] Muhammad et al., DLIME: A Deterministic Local Interpretable Model-Agnostic Explanations Approach for Computer-Aided Diagnosis Systems
>
> [3] Zhengze et al., S-LIME: Stabilized-LIME for Model Explanation

---

> > ### Comment · Reviewer_jFSL · 2023-08-14
> > **Thank you for the response**
> >
> > I would like to thank the authors for their response. Although I still feel that this paper feels incomplete without the human angle, I will not fight for rejection due to this. Overall, other concerns have been satisfactorily addressed and thus I will raise my score to 5.

---

> > > ### Author Response · Authors · 2023-08-21
> > > **Human-interpretability experiment results**
> > >
> > > Thank you for your feedback and suggestions. We have incorporated human-interpretability experiments as you recommended. Our experimental design is as follows:
> > >
> > > * Our experiment consists of two parts:
> > >
> > >   1. Selecting images where the model's predictions are correct and presenting the original images along with explanations from LIME and GLIME to the participants. The participants are asked to rate the degree of matching between the explanations provided by the algorithms and their intuitive understanding on a scale of 1-5, where 1 indicates a significant mismatch and 5 indicates a strong match.
> > >
> > >   2. Selecting images where the model's predictions are incorrect and presenting the original images along with explanations from LIME and GLIME to the participants. The participants are asked to rate the degree of help provided by the explanations in understanding the model's behavior and identifying the reasons for the incorrect predictions on a scale of 1-5, where 1 indicates no help at all and 5 indicates significant help.
> > >
> > > * Each part consists of ten randomly selected images.
> > >
> > > * The participants in the experiment are college students from diverse backgrounds with no prior knowledge of machine learning. There are ten participants for each part, and the participants for the two parts are different.
> > >
> > > Here are the results of the experiment:
> > >
> > > * When participants were shown images where the model's predictions were correct, along with explanations from LIME and GLIME, they gave an average score of 2.96 to LIME and an average score of 3.37 to GLIME. Overall, GLIME had an average score 0.41 higher than LIME. For seven out of the ten images, GLIME had a higher average score than LIME. We performed a t-test on the scores of LIME and GLIME, resulting in a t-value of -1.36 and a p-value of 0.29.
> > >
> > > * When participants were shown images where the model's predictions were incorrect, along with explanations from LIME and GLIME, they gave an average score of 2.33 to LIME and an average score of 3.42 to GLIME. GLIME had an average score 1.09 higher than LIME for all ten images. We performed a t-test on the scores of LIME and GLIME, resulting in a t-value of -8.75 and a p-value of $1.08\times 10^{-5}$.
> > >
> > > The results of the second part of the experiment indicate that GLIME is more effective in helping people understand the model's behavior and debug the model. Although the results of the first part of the experiment are not statistically significant, they still reflect the relative advantage of GLIME over LIME to some extent. We will conduct more experiments to obtain statistically significant results and include results in the final version. We hope that our current experimental results address your concerns.

---

> > > > ### Comment · Reviewer_jFSL · 2023-08-21
> > > > **Thank you for all your hardwork**
> > > >
> > > > My concerns stand resolved and I am raising my rating to 7.
> > > > I would like to thank the authors for their hardwork and good luck with the paper.

---

> > > > > ### Author Response · Authors · 2023-08-21
> > > > > **Thank you!**
> > > > >
> > > > > Thank you for your prompt response. We are glad that we were able to address your concerns. We appreciate the time and effort you have put into reviewing our paper. Wishing you a pleasant day!

---

### Official Review · Reviewer_sX7G · 2023-07-01

**Soundness:** 4 excellent
**Presentation:** 4 excellent
**Contribution:** 4 excellent
**Rating:** 7
**Confidence:** 3

**Summary:**

This paper proposes a new explanation method, GLIME, which provides more general, stable and local LIME explanations over ML models. Specifically, the authors demonstrate that small sample weights cause the instability of LIME, which results in dominance of regularization slow convergence and worse local fidelity.To address those issues, the authors proposed GLIME framework, which takes a slightly different form of LIME. Through rigorous theoretical analysis and some experiments, the authors can demonstrate that GLIME addressed the above issues and outperformed LIME.

**Strengths:**

+ The authors addressed a very important problem, i.e., the well-known instability issue of LIME, and proposed an effective solution to address it.
+ The authors conducted a rigorous theoretical analysis to support their claims, which is very convincing.
+ The overall presentation is very clear and easy to follow.

**Weaknesses:**

+ The experiments are only conducted on one dataset, i.e., ImageNet dataset. It would be better if the authors could show more results on more benchmark datasets
+ Some properties that are studied in theory for GLIME are not empirically verified. For example, in Section 4.1, the authors showed that their method can converge faster than LIME. Although they provide clear proof for it, the authors did not demonstrate it in experiments. So it would be better if some empirical experiments can cover this.

**Questions:**

+ How about the experimental results of using GLIME on other datasets?
+ How about the empirical comparison between GLIME and LIME in terms of the convergence speed?

**Limitations:**

Not applicable.

---

> ### Author Rebuttal · Authors · 2023-08-09
>
> We thank Reviewer sX7G for reviewing our paper and for the insightful comments. We hope our answers will address your concerns.
>
> > **Q1:** The experiments are only conducted on one dataset, i.e., ImageNet dataset. It would be better if the authors could show more results on more benchmark datasets.
>
> **A1:** We have also conducted experiments on text data, utilizing the DistilBERT model. We select 100 data points from the IMDb dataset as inputs for explanation. In our experiments, we compare the performance of GLIME-Binomial and LIME, and the Jaccard Index results are presented in the table below. Our findings indicate that GLIME-Binomial exhibits significantly higher stability than LIME across various values of $\sigma$ and sample sizes. Particularly, when $\sigma$ is small, GLIME-Binomial demonstrates a substantial improvement in stability compared to LIME.
>
> GLIME-B: GLIME-Binomial;
>
> | # samples | $\sigma=0.25$ |       | $\sigma=0.5$ |       | $\sigma=1$ |       | $\sigma=5$ |       |
> | --------- | ------------- | ----- | ------------ | ----- | ---------- | ----- | ---------- | ----- |
> |           | GLIME-B       | LIME  | GLIME-B      | LIME  | GLIME-B    | LIME  | GLIME-B    | LIME  |
> | 258       | 0.533         | 0.406 | 0.533        | 0.407 | 0.533      | 0.421 | 0.533      | 0.509 |
> | 512       | 0.607         | 0.414 | 0.607        | 0.417 | 0.607      | 0.441 | 0.607      | 0.567 |
> | 1024      | 0.679         | 0.431 | 0.683        | 0.437 | 0.683      | 0.480 | 0.683      | 0.637 |
>
> The $R^2$ values for GLIME and LIME are presented in the table below. The sample size is fixed at 1024. The results indicate that GLIME exhibits superior local fidelity compared to LIME, particularly when $\sigma$ is small.
>
> GLIME-B: GLIME-Binomial;
>
> |       | $\sigma=0.25$ |       | $\sigma=0.5$ |       | $\sigma=1$ |       | $\sigma=5$ |       |
> | ----- | ------------- | ----- | ------------ | ----- | ---------- | ----- | ---------- | ----- |
> |       | GLIME-B       | LIME  | GLIME-B      | LIME  | GLIME-B    | LIME  | GLIME-B    | LIME  |
> | $R^2$ | 0.688         | 0.001 | 0.691        | 0.160 | 0.693      | 0.579 | 0.693      | 0.682 |
>
> We will conduct more experiments in the future and incorporate the results into the final version.
>
> > **Q2:** Some properties that are studied in theory for GLIME are not empirically verified. For example, in Section 4.1, the authors showed that their method can converge faster than LIME. Although they provide clear proof for it, the authors did not demonstrate it in experiments. So it would be better if some empirical experiments can cover this.
>
> **A2:** The comparison is presented in Figure 4(a). In Figure 4(a), we present the stability of GLIME and LIME with respect to random seeds and parameters. GLIME-Binomial, which is equivalent to LIME, is much more stable than LIME, especially when $\sigma$ is small. Under default setting where $\sigma=0.25$, GLIME requires 256 sample to have top-20 Jaccard Index $\approx 0.9$ while with 2048 samples, LIME only has top-20 Jaccard Index$\approx 0.7$.  This is an empirical evidence that GLIME converges much faster than LIME. Please refer to Section 5.1 and Figure 4(a) for more details.

---

> > ### Comment · Reviewer_sX7G · 2023-08-12
> > **Thanks for the authors' efforts**
> >
> > I think the authors addressed my concerns. I also read other reviewers' comments and I think overall the responses look good to me. So I raised my score. But I would love to communicate with other reviewers if any other significant issues are raised.

---

> > > ### Author Response · Authors · 2023-08-14
> > >
> > > Thank you for your response and encouraging feedback. We are glad that we are able to help address your concerns.

---

### Official Review · Reviewer_DXPw · 2023-07-05

**Soundness:** 4 excellent
**Presentation:** 3 good
**Contribution:** 3 good
**Rating:** 7
**Confidence:** 4

**Summary:**

In this paper, the authors present GLIME an approach for explainable ai that generalizes LIME. Here, the authors present a framework that encompasses different explainability methods as instantiations of different aspects such as loss function, sampling function, etc.

The authors also present an analysis of problematic cases for LIME. More precisely, they show how the interaction of the weighting and regularization can cause instability in the explanations and how the samples generated in LIME might not be close to the original sampling space.

The paper then presents different sampling procedures and show empirically how they converge and how stable the explanations are given different parameterizations.


**Strengths:**

I find the paper insightful, in particular the aspect of the weights becoming all zeros in the standard case for low values of sigma.

The paper is easy to read, technically sound and guides the reader through the concepts in a solid yet understandable way.
The technical contributions are solid and overall provides a good foundation for further research.

Overall the paper is original and I would say significant as it has the potential to become the standard replacement for LIME.

**Weaknesses:**

The main concern I have is regarding the empirical section. In particular, you mention two main issues with LIME being the interaction of the weighting with regularization and sub-par sampling. However, it would seem like ALIME does not suffer from those two issues. It would be good to see a comparison of GLIME and ALIME in Fig. 4.

There are some minor improvements I would suggest on the presentation.
I would suggest you unify the color scheme in Fig. 4 and if possible present as many of the methods in both graphs.
Is there a typo in the norm of the weighting function in line 171? shouldn't it be 2 and 2?
The language on the sub-section in Feature attributions could be improved.


**Questions:**

My main concern is regarding the contribution and the comparison to ALIME. Could you mention a bit more in terms of your contributions regarding that method?
This does not diminish your theoretical contributions in terms of the convergence in comparison to LIME, however it can potentially make GLIME less appealing to the general user than ALIME.

Regarding your GLIME-gauss, and I believe ALIME is similar, wouldn't the sampling space be too close to the original image we want to explain? It seems like simply a noise addition similar to difussion models.

**Limitations:**

I don't consider the paper to have potential negative societal impact.

---

> ### Author Rebuttal · Authors · 2023-08-09
>
> We thank Reviewer DXPw for reviewing our paper and for the insightful comments. We hope our answers will address your concerns.
>
> > **Q1:** You mention two main issues with LIME being the interaction of the weighting with regularization and sub-par sampling. However, it would seem like ALIME does not suffer from those two issues. It would be good to see a comparison of GLIME and ALIME in Fig. 4.
>
> **A1:** Although ALIME improves stability and local fidelity over LIME, GLIME still outperforms ALIME. One major difference between ALIME and LIME is that ALIME uses an encoder to encode samples into embedding space and compute their distance with the explained input in embedding space $\\\|\mathcal{AE}(\mathbf{z}) - \mathcal{AE}(\mathbf{x})\\\|_2$ while LIME uses a binary vector $\mathbf{z} \in \\\{0,1\\\}^d$ to represent a sample, and use $\\\|\mathbf{1} - \mathbf{z}\\\|_2$ as the distance between the sample and the explained input. ALIME use distance in embedding space to weight samples. Therefore, if the samples generated by ALIME is distant from $\mathbf{x}$, sample weights may still be very small and cause instability problem.
>
> We conduct experiments to compare GLIME and ALIME. We utilize the VGG16 model provided by the repository https://github.com/Horizon2333/imagenet-autoencoder as the encoder in ALIME for our experiments on ImageNet. The table below shows the results of the experiments. It can be observed that the improvement of ALIME is still not as significant as that of GLIME, especially when $\sigma$ is small or the sample size is small. We will include results of ALIME in Figure 4 in the final version.
>
> GLIME-B: GLIME-Binomial;  GLIME-G: GLIME-Gauss
>
> | # samples | $\sigma=0.25$ |         |       | $\sigma=0.5$ |         |       | $\sigma=1$ |         |       | $\sigma=5$ |         |       |
> | --------- | ------------- | ------- | ----- | ------------ | ------- | ----- | ---------- | ------- | ----- | ---------- | ------- | ----- |
> |           | GLIME-B       | GLIME-G | ALIME | GLIME-B      | GLIME-G | ALIME | GLIME-B    | GLIME-G | ALIME | GLIME-B    | GLIME-G | ALIME |
> | 128       | 0.952         | 0.872   | 0.618 | 0.596        | 0.875   | 0.525 | 0.533      | 0.883   | 0.519 | 0.493      | 0.865   | 0.489 |
> | 256       | 0.981         | 0.885   | 0.691 | 0.688        | 0.891   | 0.588 | 0.602      | 0.894   | 0.567 | 0.545      | 0.883   | 0.539 |
> | 512       | 0.993         | 0.898   | 0.750 | 0.739        | 0.904   | 0.641 | 0.676      | 0.908   | 0.615 | 0.605      | 0.898   | 0.589 |
> | 1024      | 0.998         | 0.911   | 0.803 | 0.772        | 0.912   | 0.688 | 0.725      | 0.915   | 0.660 | 0.661      | 0.910   | 0.640 |
>
> > **Q2:** Could you mention a bit more in terms of your contributions regarding that method?  Regarding your GLIME-gauss, and I believe ALIME is similar, wouldn't the sampling space be too close to the original image we want to explain? It seems like simply a noise addition similar to diffusion models.
>
> **A2:** Both GLIME-Gauss and ALIME utilize Gaussian sampling and employ Ridge regression to solve linear explanations. However, there are notable differences between the two methods.
>
> * Firstly, ALIME initially trains an auto-encoder and then uses this auto-encoder to compute embeddings for each sample. On the other hand, GLIME does not require the training of an auto-encoder or any calculation of embeddings. Consequently, GLIME is significantly more efficient than ALIME.
>
> * Secondly, in ALIME, sample weights are computed based on the distance between the embeddings of the samples and the embedding of the original input, denoted as $\mathbf{x}$. These sample weights are then used in a weighted Ridge regression to obtain explanations. Conversely, GLIME does not involve the use of sample weights. In ALIME, sample weights emphasize locality, whereas GLIME enforces locality by employing a sample distribution that more frequently samples samples closer to $\mathbf{x}$.
> * Based on the comparison of the results between GLIME and ALIME mentioned above, although ALIME has improved stability compared to LIME, GLIME shows even better stability than ALIME, especially when the value of sigma is small and the sample size is small.
>
> Additionally, GLIME-Gauss introduces Gaussian noise to super-pixels, while diffusion models add noise to pixels. The distance between samples and the original image can be controlled by adjusting the parameter $\sigma$. In Figure 3, we showcase the distances of samples from the original image, with larger $\sigma$ resulting in samples being farther from the desired $\mathbf{x}$.
>
> > **Q3:** I would suggest you unify the color scheme in Fig. 4 and if possible present as many of the methods in both graphs.
>
> **A3:** Thanks for your suggestions. In the final version, we will unify Figure 4(a) and Figure 4(b). However, it is important to clarify that these figures are intended to compare different aspects of LIME and GLIME.
>
> Figure 4(a) aims to demonstrate how GLIME improves stability compared to LIME, while also illustrating the impact of regularization and weighting on the stability of LIME. To provide a comprehensive comparison, we will include LIME with regularization parameter $\lambda$ set to 0 (LIME$+\lambda=0$), LIME with $\lambda = e^{-d/\sigma^2}$ (LIME$+\lambda = e^{-d/\sigma^2}$), and LIME with sample weight function $\pi$ set to 1 (LIME$+\pi=1$).
>
> On the other hand, Figure 4(b) focus on comparing the local fidelity of LIME and GLIME, specifically by including different sample distributions used in GLIME.
>
> > **Q4:**  Is there a typo in the norm of the weighting function in line 171? shouldn't it be 2 and 2? The language on the sub-section in Feature attributions could be improved.
>
> **A4:** Thank you for providing the correction. The accurate expression should be $\pi(\mathbf{z^\prime}) = \exp(-\\\|\mathbf{1} - \mathbf{z}^\prime\\\|_2^2 /\sigma^2)$. We will make sure to revise the language in the final version for further improvement.

---

> > ### Comment · Reviewer_DXPw · 2023-08-14
> >
> > I'm satisfied with the rebuttal provided by the authors and the discussion. Therefore I'm raising my score.

---

> > > ### Author Response · Authors · 2023-08-21
> > >
> > > Thank you for your response and encouraging feedback. We are glad that you are satisfied with our response.

---

### Official Review · Reviewer_NTaD · 2023-07-06

**Soundness:** 4 excellent
**Presentation:** 3 good
**Contribution:** 3 good
**Rating:** 7
**Confidence:** 4

**Summary:**

The paper introduces GLIME as a solution to tackle the issues of instability and diminished local fidelity encountered in the original LIME method. To address the problem of instability, GLIME employs a novel sampling scheme that guaranteed to have a faster sampling rate. The diminished local fidelity problem is resolved by modifying sampling distribution so that nearby samples have higher probability to be sampled.

Disclaimer: I only read the main text and do not check correctness of the proposed sample complexity argument.

**Strengths:**

1. The problem they tackle with is specific and well-formulated. The proposed solution is simple and effective.
2. Their methods are supported by sample complexity analysis. This analysis not only provides mathematical evidence that the original LIME approach necessitates an exponentially large number of samples to achieve convergence, but also demonstrates that their proposed method requires only a polynomial number of samples, offering a significant improvement in efficiency.

**Weaknesses:**

One weakness would be limited applicability of the proposed GLIME. The paper only demonstrates it can only be applied to the image domain. As other features from different domains, such as texts or categorical features, are not necessarily to be continuous, GLIME equipped with continuous distributions may not resolve the local fidelity issue.

**Questions:**

In Section 4.3 of the paper, it is stated that the local fidelity problem arises due to the utilization of a high regularization weight. Could this issue be addressed by reducing the regularization weight or, in more extreme cases, completely eliminating the regularization.

**Limitations:**

The potential societal impact is not stated in the main paper.

---

> ### Author Rebuttal · Authors · 2023-08-09
>
> We thank Reviewer NTaD for reviewing our paper and for the insightful comments. We hope our answers will address your concerns.
>
> > **Q1:** One weakness would be limited applicability of the proposed GLIME. The paper only demonstrates it can only be applied to the image domain. As other features from different domains, such as texts or categorical features, are not necessarily to be continuous, GLIME equipped with continuous distributions may not resolve the local fidelity issue.
>
> **A1:** We have conducted experiments on text data. We use the DistilBERT model and select 100 data points from the IMDb dataset as inputs to be explained. We run experiments comparing GLIME-Binomial and LIME, and the results are shown in the table below. The sample size is set to 1024. From the results, it can be seen that GLIME has better local fidelity compared to LIME, especially when $\sigma$ is small.
>
> GLIME-B: GLIME-Binomial;
>
> |       | $\sigma=0.25$ |       | $\sigma=0.5$ |       | $\sigma=1$ |       | $\sigma=5$ |       |
> | ----- | ------------- | ----- | ------------ | ----- | ---------- | ----- | ---------- | ----- |
> |       | GLIME-B       | LIME  | GLIME-B      | LIME  | GLIME-B    | LIME  | GLIME-B    | LIME  |
> | $R^2$ | 0.688         | 0.001 | 0.691        | 0.160 | 0.693      | 0.579 | 0.693      | 0.682 |
>
> For data with discontinuous feature values, the samples $\mathbf{z^\prime}$s naturally tend to be far away from the input being explained. However, because of the large distances, the weights of each sample after weighting are very low. This may cause the sum-of-squares term to be dominated by the regularization term, leading to explanations tending towards zero and resulting in a smaller $R^2$ and poorer local fidelity. GLIME does not suffer from this problem, as the sum-of-squares term is not dominated by the regularization term, leading to better local fidelity. We will conduct additional experiments in the future to compare GLIME and LIME.
>
> > **Q2:** In Section 4.3 of the paper, it is stated that the local fidelity problem arises due to the utilization of a high regularization weight. Could this issue be addressed by reducing the regularization weight or, in more extreme cases, completely eliminating the regularization.
>
> **A2:** In our paper, we propose that the instability issue in LIME is attributable to the use of a high regularization weight and low sample weights. To address this concern, we conducted experiments aimed at evaluating whether reducing the regularization parameter $\lambda$ or increasing the sample weight function $\pi(\cdot)$ could improve stability. For a comprehensive analysis of the results and discussion, please refer to Figure 4(a) and lines 278-289. It is important to note that while removing regularization does offer some improvement, it is not as significant as the improvement achieved by GLIME. This is because, in the absence of regularization, the LIME solution involves computing $(Z^\top WZ)^{-1}$, where $W$ is a diagonal matrix with sample weights on its diagonal. The dependence on $W$ makes LIME sensitive to sample weights. Furthermore, since the sample weights are nearly zero, $Z^\top WZ$ becomes low-rank, resulting in numerical instability. Additionally, increasing the sample weights to 1 only provides limited improvement in stability for LIME. This is because the large sampling space of LIME causes the obtained explanations to highly depend on the selected samples. GLIME, on the other hand, does not encounter these issues as its solution does not require inverting a low-rank matrix, and it operates within a local sampling space.
>
> > **Q3:** The potential societal impact is not stated in the main paper.
>
> **A3:** The positive impact of GLIME is its ability to enhance the interpretability of machine learning, making the use of machine learning in various fields safer and more trustworthy. It reduces the potential for social security incidents caused by machine learning. It can help users better understand the behavior of machine learning models. The potential negative societal impacts of GLIME are not yet clear at the moment.

---

> > ### Comment · Reviewer_NTaD · 2023-08-12
> > **Thanks for the author's reply**
> >
> > I believe the authors have taken into account the points I raised, so I raised my score to 7. Additionally, I've gone through feedback from other reviewers, and on the whole, I find the authors' responses satisfactory.

---

> > > ### Author Response · Authors · 2023-08-14
> > >
> > > Thanks for your reply and encouraging feedback. We are glad to see that you are satisfied with our responses.

---

### Decision · Program_Chairs · 2023-09-21

**Decision:**

Accept (spotlight)

**Comment:**

All reviewers recommend acceptance and point out that the contribution is of interest to the community.